# 🎏 PATRA: Pattern-Aware Alignment and Balanced Reasoning for Time Series Question Answering

**Junkai Lu** [* 1]   **Peng Chen** [* 1]   **Xingjian Wu** [* 1]   **Yang Shu** [1]   **Chenjuan Guo** [† 1]   **Christian S. Jensen** [2]   **Bin Yang** [1]

## Abstract

Time series reasoning demands both the perception of complex dynamics and logical depth. However, existing LLM-based approaches exhibit two limitations: they often treat time series merely as text or images, failing to capture the patterns like trends and seasonalities needed to answer specific questions; and when trained on a mix of simple and complex tasks, simpler objectives often dominate the learning process, hindering the development of deep reasoning capabilities. To address these limitations, we propose the Pattern-Aware Alignment and Balanced Reasoning model (PATRA), introducing a pattern-aware mechanism that extracts trend and seasonality patterns from time series to achieve deep alignment. Furthermore, we design a task-aware balanced reward to harmonize learning across tasks of varying difficulty, incentivizing the generation of coherent Chains of Thought. Extensive experiments show that PATRA outperforms strong baselines across diverse Time Series Question Answering (TSQA) tasks, demonstrating superior cross-modal understanding and reasoning capability.

🐙 https://github.com/decisionintelligence/PATRA

## 1. Introduction

Time series data is used widely across application domains such as Energy (Tzelepi et al., 2023), AIOps (Zhong et al., 2023), Traffic (Fang et al., 2021), Weather (Nguyen et al., 2023), and Finance (He et al., 2023), characterized by complex temporal dynamics encompassing trend and seasonality patterns. The emergence of Large Language Models (LLMs) has given rise to Time Series Question Answering (TSQA),

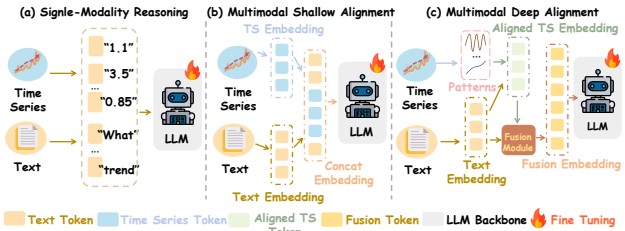

*Figure 1.* Comparison of alignment paradigms in Time Series Question Answering. (a) Single-Modality Reasoning treats time series as text sequences, converting continuous numerical points into text tokens. (b) Multimodal Shallow Alignment introduces time series as a separate modality but relies on simple concatenation of embeddings. (c) Multimodal Deep Alignment addresses these limitations by explicitly decomposing time series into distinct patterns and fusing them with text embeddings via a dedicated module, ensuring that semantic reasoning is firmly grounded in physical data behaviors.

leveraging the capabilities of LLM Backbones to analyze and reason over time series (Kong et al., 2025; Guan et al., 2025a) through natural language.

The patterns found in time series enable applications to reason about the dynamics of underlying systems, thereby facilitating decision-making. For instance, in financial trading, the strategies for buying or selling hinge on cyclic pullback dynamics, which relate to patterns that can be found in time series (Wu et al., 2023; Liu et al., 2023; Qiu et al., 2024b). However, merely relying on LLMs to analyze time series is insufficient for high-quality reasoning and decision-making. The model struggles to align temporal dynamics with query-specific semantic concepts without extracting the patterns. Consequently, a paradigm shift is needed: moving from reasoning on the raw point-wise values to potential temporal patterns, enabling decisions aligned with data reality. While recent works have tried to bridge this gap by adapting paradigms inspired by Textual QA (Luo et al., 2025; Song et al., 2025) or multimodal fusion strategies inspired by VLMs (Ntinou et al., 2025; Liao et al., 2025; Chen et al., 2025d), the barriers manifest in two key challenges:

**First, existing models fail to ground semantic reasoning in time series patterns, resulting in incomplete analysis.** Previous research approaches to TSQA employ *single-*

---

[*]Equal contribution  [1]East China Normal University, Shanghai, China [2]Aalborg University, Aalborg, Denmark. Correspondence to: Chenjuan Guo <cjguo@dase.ecnu.edu.cn>.

*Proceedings of the $43^{rd}$ International Conference on Machine Learning*, Seoul, South Korea. PMLR 306, 2026. Copyright 2026 by the author(s).

*modality temporal reasoning* as shown in Figure 1(a), treating time series essentially as text sequences (Liu et al., 2025b; Guan et al., 2025a). These methods convert numerical points directly into text tokens to leverage the reasoning capabilities of LLMs. Alternatively, studies have attempted *multimodal time series reasoning* by adapting architectures from Vision Language Models (VLMs) (Xie et al., 2025; Jin et al., 2024) as shown in Figure 1(b). While these methods introduce time series as a separate modality, they typically rely on "shallow alignment": merely projecting time series patches and concatenating them with text embeddings. This approach mimics image-text alignment without considering the specific properties of time series data. Unlike static images, time series possess multi-level dynamics; thus, simple concatenation fails to achieve "deep alignment" as shown in Figure 1(c), leaving a gap where semantic queries cannot be effectively grounded in the specific patterns of the time series.

**Second, the task heterogeneity in TSQA leads to optimization imbalances, which impede the acquisition of deep reasoning capabilities.** Unlike specialized models, a general TSQA system must handle a broad spectrum of objectives, ranging from straightforward discriminative queries to complex, open-ended generative reasoning. This results in different learning gradients and reward thresholds: simple tasks offer low effort to success, while complex reasoning demands substantial exploration. Unfortunately, existing methods fall short in addressing this dilemma. Most approaches rely primarily on Supervised Fine-Tuning (SFT), limiting their ability to generalize beyond training data. While a few recent works attempt to incorporate Reinforcement Learning (RL) (Liu et al., 2025b), they are either restricted to single-task scenarios—where cross-task variance is absent, or fail to account for heterogeneity in multi-task settings. Consequently, models tend to maximize expected returns by overfitting to easier tasks (Ruder, 2017), thereby neglecting the development of intricate temporal reasoning skills. Thus, a mechanism capable of balancing optimization difficulties across diverse tasks remains absent.

To address these challenges, we propose **PATRA**, a framework designed to improve both cross-modal understanding and reasoning in TSQA through two core innovations. First, to achieve better cross-modal understanding, we introduce a **Pattern-Aware Alignment Mechanism**. Unlike methods that rely on simple feature projection, this mechanism explicitly decomposes time series data into clear patterns, such as trend and seasonality. By directly mapping these physical patterns to the semantic representations in the text query, the model can effectively use structural data features during reasoning. This ensures that the generated answers are based on precise data details rather than general text knowledge.

Second, to enhance reasoning capabilities beyond the lim-

its of Supervised Fine-Tuning (SFT), we propose a **Reinforcement Learning-based Cross-Task Training Paradigm**. Since general TSQA involves tasks with varying difficulty levels, standard training often leads to optimization imbalance. To solve this, our key innovation is a **Task-Aware Balanced Reward** design. This mechanism adjusts reward signals according to the specific characteristics of each task. By doing so, it prevents the model from focusing only on easy tasks (reward hacking) and ensures a stable, unified optimization process that builds robust reasoning skills across diverse time series scenarios. Our contributions are summarized as follows:

- We propose a *pattern-aware alignment mechanism* for TSQA, aligning textual questions with time series patterns, bridging the gap between language and time series data to improve reasoning capabilities.

- We propose an RL-enhanced training paradigm that complements supervised fine-tuning (SFT), optimizing the model across diverse TSQA formats, enabling stronger reasoning and cross-task generalization.

- Extensive experiments show that PATRA outperforms strong baselines in TSQA tasks, confirming the effectiveness of our model in time series understanding and reasoning.

## 2. Related Works

**Multimodal LLMs.** Multimodal large language models (MLLMs) have rapidly advanced in recent years, enabling the integration of diverse modalities such as images (Kurtic et al., 2025), videos (Zhang et al., 2025d), audio (Dua et al., 2025), and graphs (Guan et al., 2025b) for unified understanding and reasoning (Yin et al., 2024). By aligning heterogeneous representations, these models fully leverage the language understanding and reasoning capabilities of LLMs, achieving strong results in tasks such as visual question answering and multimodal reasoning (Li et al., 2023; Alayrac et al., 2022). Beyond performance improvements, MLLMs also highlight the importance of cross-modal understanding. This demonstrates that equipping LLMs with the ability to handle structured signals beyond natural language can significantly broaden their applicability. Motivated by these advances, researchers have started to investigate whether similar multimodal modeling and reasoning paradigms can be extended to the domain of time series (Hu et al., 2025; Liu et al., 2025a), which pose unique challenges due to their numerical, dynamic, and non-stationary.

**Time series Reasoning.** Inspired by MLLMs, recent studies have attempted to combine time series with LLMs. Early works such as *TimeLLM* (Jin et al., 2024), *GPT4TS* (Zhou et al., 2023), and *LLMTime* (Nate Gruver & Wilson, 2023)

focus on concatenating temporal and textual tokens, fine-tuning on large-scale time series corpora, or exploring zero-shot prediction. However, they lack a unified framework for modeling, reasoning, and natural language interaction with time series. More recent approaches, including *ChatTS* (Xie et al., 2025), Time-MQA (Kong et al., 2025), *IT-former* (Wang et al., 2025b), and TimeOmni-1 (Guan et al., 2025a), emphasize interpretability by generating reasoning processes in addition to numerical outputs. Despite these efforts, existing methods still struggle with alignment between temporal features and textual queries, and cross-task generalization (Ruder, 2017; Amodei et al., 2016).

## 3. Preliminaries

Time series reasoning covers a variety of tasks, such as trend inference, pattern recognition, et al. To unify these tasks, we propose a paradigm based on *Time Series Question Answering* (TSQA), reorganizing all time series tasks in the form of question answering, thereby constructing a unified processing framework. Formally, given a time series reasoning problem $\mathcal{Q}$ and its time series set $\mathcal{S} = \{\mathcal{S}_1, \mathcal{S}_2, \ldots, \mathcal{S}_{\mathrm{M}}\}$, where the length of each $\mathcal{S}_i$ may not be fixed, the goal of the model is to generate the corresponding natural language answer $\mathcal{A}$ based on these two inputs. The entire task can be formally represented as follows:

$$\mathcal{A} = f(\mathcal{Q}, \mathcal{S}), \tag{1}$$

where the function $f(\cdot)$ can be implemented as an LLM, or an MLLM. For the MLLM, the specific form can be further expressed as:

$$f(\mathcal{Q}, \mathcal{S}) = g(\mathcal{Q}, \Psi(\mathcal{S})). \tag{2}$$

Here, $g(\cdot)$ represents an MLLM, $\Psi(\cdot)$ represents a feature extraction for time series, thereby mapping the original time series to a representation space suitable for cross-modal alignment and reasoning.

## 4. Methodology

We propose **PATRA**, a method based on *Pattern-Aware Alignment* and *RL-Augmented Training*, combined with a composite reward for multiple task formats. The architecture of our proposed model, **PATRA**, is shown in Figure 2. The text and time series are first characterized by their respective modality encoders, then the two modalities are aligned through pattern-aware alignment to reduce the gap between them, and finally, the final result is given through reasoning by the LLM backbone.

### 4.1. Pattern-Aware Alignment

There is a fundamental discrepancy between the two modalities: natural language is a discrete and hierarchical symbolic sequence, while time series data are continuous numerical data. This leads to substantial differences in semantic granularity and information density, making it challenging to align them directly. To address this, we propose a *Pattern-Aware Alignment Module*. Unlike existing methods that align only to one time series pattern, it integrates three types of time series patterns, ensuring a deeper reasoning process, thereby capturing cross-modal relationships at multiple semantic levels. It is mainly divided into four parts: *Modality Encoding*, *Latent Decomposition*, *Text Extraction*, and *Pattern-Aware Alignment Mechanism*.

**Modality Encoding.** Effective cross-modal reasoning over time series question answering relies on capturing rich semantics from both natural language queries and time series data. To achieve this, we need representations that encapsulate intrinsic modality-specific features while facilitating robust cross-modal interaction. For the textual query $\mathcal{Q}$, we first tokenize it and embed each token into the LLM's embedding space:

$$\mathbf{X}_{\mathrm{text}} = \mathrm{Embed}(\mathrm{Tokenizer}(\mathcal{Q})), \tag{3}$$

where $\mathbf{X}_{\mathrm{text}} \in \mathbb{R}^{L \times d}$, $L$ is the number of tokens, and $d$ is the embedding dimension. These pretrained embeddings, derived from the LLM's vocabulary, encapsulate rich syntactic and semantic knowledge learned from large-scale text corpora.

For the input multivariate time series set $\mathcal{S} = \{\mathcal{S}_1, \ldots, \mathcal{S}_M\}$, we first pad all series to the same length to get the padded token $\mathbf{X}_{\mathrm{pad}}$, normalize with Instance Normalization (Kim et al., 2022) to mitigate distributional shifts, and then perform patching (Cirstea et al., 2022; Nie et al., 2023) and embedding to obtain temporal tokens:

$$\begin{aligned} \mathbf{X}_{\mathrm{patch}} &= \mathrm{Patching}(\mathbf{X}_{\mathrm{pad}}), \\ \mathbf{X}_{\mathrm{ts}} &= \mathrm{Embedding}(\mathbf{X}_{\mathrm{patch}}), \end{aligned} \tag{4}$$

where $\mathbf{X}_{\mathrm{ts}} \in \mathbb{R}^{(M \cdot N) \times d}$, $N$ is the number of patches per series, $d$ is the embedding dimension, and $M$ is the number of time series. This process aggregates information across multiple time points, yielding rich temporal features suitable for subsequent pattern-aware alignment and reasoning.

**Latent Decomposition.** Instead of operating on raw numerical values, we perform decomposition within the high-dimensional embedding space to capture intrinsic temporal dynamics at a semantic level. Inspired by (Qiu et al., 2025c; Wu et al., 2021), this module disentangles the time series embeddings into three distinct latent components: full component, trend component, and seasonal component. Specifically, since the global context is encapsulated during the *Modality Encoding*, we utilize the embedding $\mathbf{X}_{\mathrm{ts}}$ as the holistic full temporal representation $\mathbf{X}_{\mathrm{ts}}^{\mathrm{f}}$. We adopt average pooling to extract the trend, acting as a moving average filter

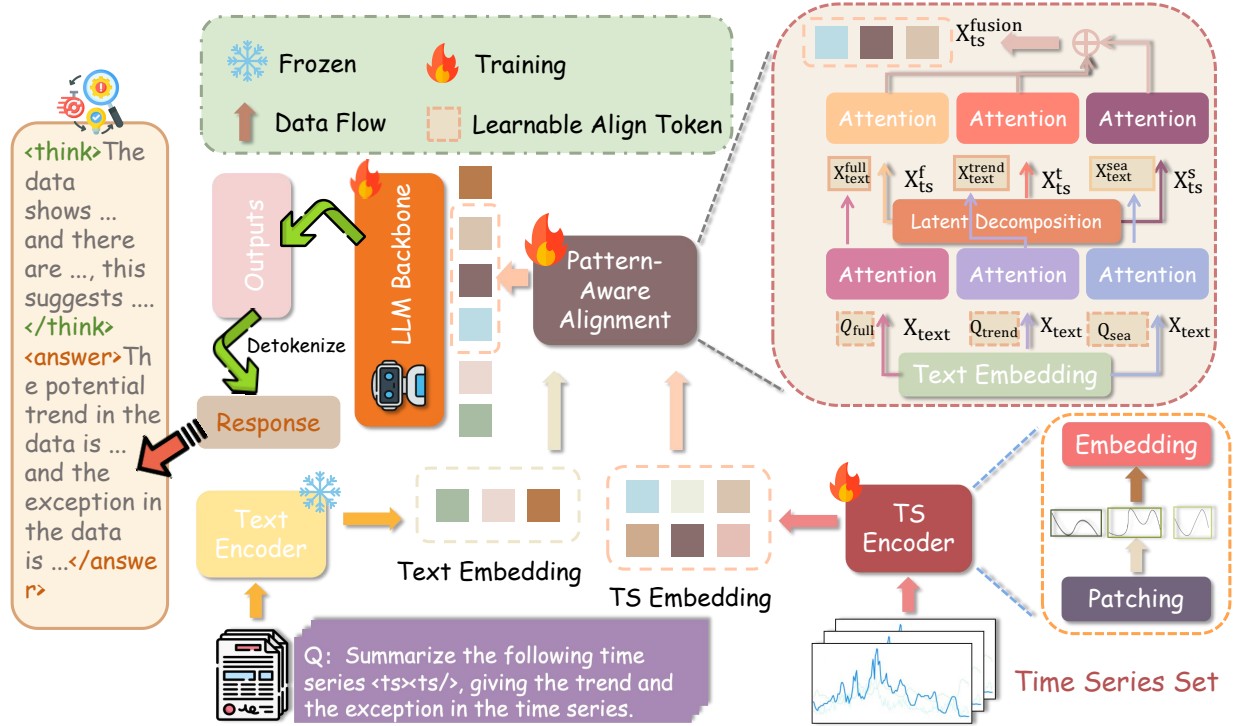

*Figure 2.* The framework of PATRA, which contains a Text Encoder for obtaining text embedding from pre-trained representations, a TS Encoder for embedding the time series, a Pattern-Aware Alignment to align the gap between text embedding and time series embedding, and an LLM Backbone to reason the questions.

that dampens noise to capture the trends. Accordingly, the latent extraction process is formulated as follows:

$$\begin{aligned}
\mathbf{X}_{ts}^{t} &= \text{Avgpool}(\text{padding}(\mathbf{X}_{ts})), \\
\mathbf{X}_{ts}^{s} &= \mathbf{X}_{ts} - \mathbf{X}_{ts}^{t},
\end{aligned} \tag{5}$$

where $\mathbf{X}_{ts}^{t}, \mathbf{X}_{ts}^{s} \in \mathbb{R}^{(M \cdot N) \times d}$ denote the extracted trend and seasonal patterns (Wu et al., 2021). To verify that Latent Decomposition preserves the original patterns of the time series, we provide a visualization analysis in Section 5.10.

**Text Extraction.** To better align with time series tokens at a fine-grained level, we first extract the text. We use three types of *Learnable Alignment tokens* (LATs) to extract the text embedding $\mathbf{X}_{full}^{text}$, $\mathbf{X}_{trend}^{text}$, and $\mathbf{X}_{sea}^{text}$. The implementation is as follows:

$$\mathbf{X}_{k}^{text} = \text{Attention}(\mathbf{Q}_{k}, \mathbf{K}, \mathbf{V}), k \in \{\text{full, trend, sea}\} \tag{6}$$

where $\mathbf{K}, \mathbf{V}$ denote the original text embeddings $\mathbf{X}_{text}$, and $\mathbf{Q}_{full}, \mathbf{Q}_{trend}, \mathbf{Q}_{sea} \in \mathbb{R}^{T \times d}$ are three learnable query tokens that extract text features and perform alignment with the temporal tokens and $\text{Attention}(\cdot)$ is implemented as a *Standard Multi-Head Attention* (Vaswani et al., 2017). Here, $T$ denotes the number of LATs used to bridge and align the textual and time series tokens and $\mathbf{X}_{full}^{text}, \mathbf{X}_{trend}^{text}, \mathbf{X}_{sea}^{text} \in \mathbb{R}^{T \times d}$ represent the three textual pattern tokens extracted by

LATs, which are subsequently aligned with the corresponding time series tokens.

**Pattern-Aware Alignment Mechanism.** After finishing *Text Extraction* and *Time Series Decomposition*, we obtain fine-grained tokens. These token sets collectively serve as the unified input for subsequent cross-modal reasoning.

Based on these, we design a *Pattern-Aware Alignment Mechanism*: interacting the three sets of textual tokens with the corresponding time series tokens respectively, enabling the time series tokens to integrate textual semantics, thereby achieving precise alignment across modalities. This mechanism explicitly establishes the semantic correspondence between textual and time series tokens. Formally, the alignment process can be expressed as follows:

$$\mathbf{Q}_{i}^{inter} = [\mathbf{X}_{i}^{text}; \mathbf{X}_{ts}^{j}], (i, j) \in \{(\text{full}, f), (\text{trend}, t), (\text{sea}, s)\} \tag{7}$$

where, we first place the extracted text token before the time series token to form three new queries, where $\mathbf{Q}_{full}^{inter}, \mathbf{Q}_{trend}^{inter}, \mathbf{Q}_{sea}^{inter} \in \mathbb{R}^{(T + M \cdot N) \times d}$. Then we employ a self-attention mechanism to enable the interaction between textual tokens and time series tokens, thereby allowing time series tokens to be integrated into textual semantics and achieving more precise alignment. Finally, we extract the time series token that has been integrated into the textual

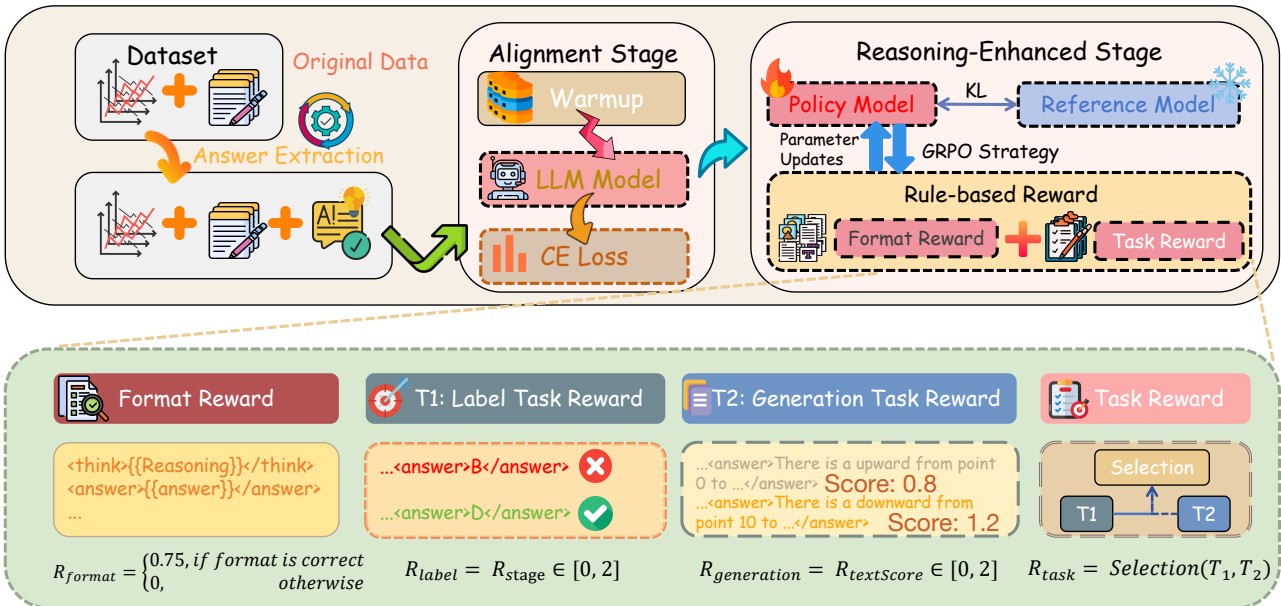

*Figure 3.* Overview of PATRA's two training stages. Alignment Stage performs supervised fine-tuning (SFT) with cross-entropy (CE) loss, while Reasoning-Enhanced Stage applies reinforcement learning, including format and task reward under the GRPO strategy to optimize the policy model.

semantics:

$$\mathbf{X}_k^{\text{inter}} = \text{SelfAttention}(\mathbf{Q}_k^{\text{inter}}), k \in \{\text{full, trend, sea}\}$$
$$\mathbf{X}_{\text{ts}}^{\text{fusion}} = \text{Fusion}(\mathbf{X}_{\text{full}}^{\text{inter}}, \mathbf{X}_{\text{trend}}^{\text{inter}}, \mathbf{X}_{\text{sea}}^{\text{inter}}), \quad (8)$$

where $\text{SelfAttention}(\cdot)$ operation is implemented as a *Standard Multi-Head Attention*. This design allows the attention mechanism to align textual and temporal information simultaneously, as each combined query can attend to relevant temporal patterns while maintaining contextual textual semantics, and $\text{Fusion}(\cdot)$ represents the fusion process of the tokens implemented as the average process and $\mathbf{X}_{\text{full}}^{\text{inter}}, \mathbf{X}_{\text{trend}}^{\text{inter}}, \mathbf{X}_{\text{sea}}^{\text{inter}}, \mathbf{X}_{\text{ts}}^{\text{fusion}} \in \mathbb{R}^{(M \cdot N) \times d}$.

As shown in Figure 2, instead of directly concatenating time series embeddings, we introduce <ts><ts/> placeholders into the textual input. This preserves the natural language structure that the LLM has been trained on, avoiding distribution shift and enabling seamless integration of temporal features. After going through the *Pattern-Aware Alignment Mechanism*, we obtain the time series token aligned with the textual modality, and then replace it with the placeholder token to obtain the complete multimodal token input:

$$\mathbf{X}_{\text{m}} = \text{TokenReplace}(\mathbf{X}_{\text{text}}, \mathbf{X}_{\text{ts}}^{\text{fusion}}), \quad (9)$$

where $\text{TokenReplace}(\cdot)$ denotes the replacement process for the token, $\mathbf{X}_{\text{m}} \in \mathbb{R}^{(L+M \cdot N - 2M) \times d}$, then we take it as the input of the *LLM Backbone* and let the LLM reason to obtain the final response:

$$\mathbf{O} = \text{LLMBackbone}(\mathbf{X}_{\text{m}}),$$
$$response = \text{DeTokenizer}(\mathbf{O}). \quad (10)$$

## 4.2. RL-Augmented Training

In scenarios involving multiple task types, such as text generation and selection-based judgment, existing LLMs for Time Series Question Answer are typically optimized with single-type task rewards, which hinders the learning of shared representations across tasks and may cause reward imbalance or reward hacking (Sun et al., 2025).

To address this, we adopt an RL-Augmented Training strategy: first, aligning LLMs enables the model to leverage time series patterns during reasoning, and then enhancing their reasoning ability through reinforcement learning. The full pipeline of training is illustrated in Figure 3.

### 4.2.1. ALIGNMENT STAGE

In the *Alignment Stage*, PATRA undergoes standard supervised fine-tuning (SFT) using cross-entropy loss, guiding the model to predict the correct answers. Through this stage, the model can gradually learn the dynamic time series patterns, which makes the model build a deeper understanding of the time series, providing a robust foundation for the subsequent reasoning stage.

### 4.2.2. REASONING-ENHANCED STAGE

For high-quality inference outputs, there are often multiple training objectives: structural correctness, task accuracy, etc. To meet these multifaceted demands, we construct a composite reward to include all these requirements simul-

taneously. It consists of the following two major rewards: format rewards and task rewards, as shown in Figure 3.

**Format Reward.** Formatting rewards are used to force the model to follow a predefined output structure. Unlike the traditional scoring only when all marks are correct, we adopt a partial reward strategy $r^{\text{format}} = r_{\text{step}}(response)$: a corresponding reward is given for each pair of valid markers identified, thereby avoiding the model from receiving no feedback due to format errors in the early stage of training. Furthermore, if the label pair appears repeatedly, a penalty will be imposed, thereby ensuring the model maintains a consistent output throughout the training process.

**Balanced Task Reward.** To address the task heterogeneity where objectives range from straightforward discrimination to open-ended generation, we design distinct reward mechanisms specifically tailored to the difficulty and nature of each task. We mainly divide them into two categories: one is the selection and judgment tasks with labels, and the other is the text generation tasks. For *labeled tasks*, instead of assigning a single binary reward based solely on the final output, we design a stage-wise reward function that evaluates the answer through successive checks. Specifically, for each generated response, we first extract the content within the <answer> tag, then assess it in stages, for example, verifying whether it falls within the candidate range and subsequently judging its correctness. Each stage contributes a partial reward. Formally, the reward is defined as:

$$r^{\text{label}} = \sum_{k=1}^{K} \lambda_k \cdot r_k(answer), \qquad (11)$$

where $r_k(\cdot)$ denotes the reward assigned at stage $k$, $K$ is the number of evaluation stages, and $\lambda_k$ is a weighting coefficient. This design provides more informative feedback without requiring ground-truth reasoning trajectories, thereby stabilizing training and improving reasoning accuracy.

For *text generation tasks*, we first extract the answers, then compare them with the given standard answers, and reward them based on the Rouge-L (Lin, 2004) score. Its reward function can be expressed as:

$$r^{\text{generation}} = \text{TextScore}(answer, y^*). \qquad (12)$$

Here, $answer$ represents the response generated by the model, $y^*$ represents the standard answer, and $\text{TextScore}(\cdot)$ implements the Rouge-L calculated between the generated response and the standard answer. This design encourages the model to generate responses that better align with the reference answer at the sequence and structural level, rather than relying solely on isolated keyword matching.

The task reward $r^{\text{task}}$ is determined by the task type: we apply the stage-wise reward for labeled tasks and the generation reward for text generation tasks. However, a crit-

ical challenge arises from the inherent optimization imbalance: simple discriminative tasks offer "low effort to success", whereas complex reasoning demands substantial exploration. Without intervention, the model tends to overfit to easier tasks to maximize expected returns ((Sun et al., 2025)), neglecting the development of intricate temporal reasoning skills. To resolve this dilemma, we map the value of both type rewards to the unified range of $[0, 2]$. This normalization bridges the gradient gap between heterogeneous tasks, ensuring the stability of gradient updates during cross-task training. More details are shown in Section 5.9 and Appendix E.

**Optimization for Reasoning.** We adopt *Group Relative Policy Optimization* (GRPO) (Shao et al., 2024; DeepSeek-AI et al., 2025) to optimize the model, leveraging group-wise normalization to handle heterogeneous reward distributions. Its optimization objective can be formalized as:

$$\mathcal{L}(\theta) = \mathbb{E}_{\tau \sim \pi_{\theta_{\text{old}}}} \left[ \frac{\pi_\theta(\tau)}{\pi_{\theta_{\text{old}}}(\tau)} \cdot \hat{A}_{\text{group}}(\tau) \right]. \qquad (13)$$

The dominance estimate $\hat{A}_{\text{group}}(\tau)$ based on the standardization of group statistics is defined as:

$$\hat{A}_{\text{group}}(\tau) = \frac{r(\tau) - \mu}{\sigma + \epsilon}, \quad r(\tau) = r^{\text{format}}(\tau) + r^{\text{task}}(\tau). \qquad (14)$$

Here, $r(\tau)$ represents the total reward of this sequence, $\mu$ and $\sigma$ are the mean and standard deviation of the rewards within the current group, and $\epsilon$ is the numerical stability constant. This method suppresses reward fluctuations, stabilizes training, and preserves relative rankings, helping the model better distinguish high and low-quality outputs.

## 5. Experiment

### 5.1. Datasets

We conduct experiments on the *TSQA* dataset (Kong et al., 2025). *TSQA* introduces a natural language–based reasoning paradigm, bridging classical time series analysis and LLM reasoning, and contains ~200,000 samples across 12+ domains. The data is split into training and test sets, with the test set divided into four tasks: **Comprehension**, **Recognition**, **Reasoning**, and **Prescience** tasks, enabling comprehensive evaluation in temporal reasoning scenarios, and the details are shown in the Appendix A.

### 5.2. Setups

**Metric.** We adopt type-specific metrics to evaluate model performance on the four tasks. For text generation tasks, we follow (Wang et al., 2025b) to use **Rouge-L** (Lin, 2004) to measure the semantic overlap between generated and

*Table 1.* Evaluation results for the four tasks, with the best and second-best performances highlighted respectively in open-source model in **red** with bold and orange with underline. We regard GPT-4o as the upper bound, which is not in the comparison set.

| Modalities | Models | Comp. | | Recog. | | Reason. | | Presc. | |
|---|---|---|---|---|---|---|---|---|---|
| | | Acc. ↑ | Rou. ↑ | Acc. ↑ | Rou. ↑ | Acc. ↑ | Rou. ↑ | Acc. ↑ | Rou. ↑ |
| Textual | GPT-4o | 50.86 | 11.99 | 69.65 | 4.75 | 50.00 | 7.75 | 66.66 | 6.78 |
| | TimeOmni-1 7B | 37.93 | 14.16 | 41.93 | 9.10 | 42.56 | 12.78 | 42.59 | 10.40 |
| | Deepseek-R1 7B | 40.52 | 13.65 | 12.41 | 8.40 | 18.24 | 10.34 | 14.81 | 8.59 |
| | Llama3 8B | 33.62 | 5.15 | 36.27 | 4.11 | 27.02 | 5.26 | 25.92 | 3.22 |
| | Qwen2.5 7B | 42.24 | _18.77_ | _45.51_ | 10.32 | _36.48_ | _18.72_ | 26.85 | 10.67 |
| | backbones | | | | *TimeMQA* | | | | |
| | Mistral 7B | 18.96 | 9.63 | 26.20 | 6.15 | 12.83 | 9.36 | 28.70 | 6.28 |
| | Llama3 8B | 28.45 | 9.60 | 36.27 | 6.69 | 22.97 | 8.86 | 40.74 | 5.63 |
| | Qwen2.5 7B | 10.34 | 17.92 | 24.41 | 14.51 | 13.51 | 16.89 | 26.85 | _15.53_ |
| Textual + Temporal | ITFormer 7B | 40.52 | 14.24 | 45.24 | _14.61_ | 30.40 | 15.58 | _44.44_ | 15.25 |
| | ChatTS 7B | _44.83_ | 13.30 | 36.00 | 13.23 | 22.97 | 15.84 | 25.92 | 13.99 |
| | **PATRA 7B** | **56.03** | **25.67** | **64.69** | **25.46** | **44.59** | **27.36** | **52.78** | **27.06** |

reference answers. For the questions with fixed labels, we report ***Accuracy*** as the proportion of correct answers.

**Evaluation Prompt.** To ensure fair evaluation, we use ***identical prompts*** across all models to eliminate performance differences introduced by prompt variation. Additionally, we set ***doSample=False*** to disable sampling, ensuring deterministic outputs and reducing randomness in model responses, improving the reliability and comparability of results. The full evaluation prompt is shown in the Appendix B.

## 5.3. Baselines

We evaluate several models as baselines on the four tasks, including *TimeOmni-1-7B* (Guan et al., 2025a), *ITFormer-7B* (Wang et al., 2025b), *DeepSeek-R1-7B* (DeepSeek-AI et al., 2025), *Llama3-8B* (Grattafiori et al., 2024), *Qwen2.5-7B* (Yang et al., 2025), TimeMQA series models (*Mistral-7B* (Jiang et al., 2023), *Llama3-8B*, and *Qwen2.5-7B* as backbone), *ChatTS-7B* (Xie et al., 2025), GPT-5.2 (OpenAI, 2025), Gemini-3-Pro (Google DeepMind, 2025) and GPT-4o (OpenAI, 2023). Our method is built upon the Qwen2.5 7B backbone. The training is performed on 4 A800 GPUs.

## 5.4. Main Results

Table 1 presents the overall performance of all baseline models and our PATRA model on four task types. Our PATRA achieves the best results across all metrics, surpassing pure text models and significantly outperforming the multimodal model ChatTS, highlighting its strong cross-modality modeling and reasoning capabilities.

In detail, PATRA achieves 56.03% accuracy on *Comprehension*, exceeding the best pure-text model by **13.79%** and improving over ChatTS by **11.20%**. On *Recognition*, it reaches 64.69%, outperforming the best pure-text model by **19.18%** and ChatTS by **28.69%**. For *Reasoning*, our

method achieves 44.59%, yielding a **21.62%** improvement over ChatTS and substantially surpassing all textual baselines. Finally, on *Prescience*, it attains 52.78%, beating the best textual model and improving over ChatTS by **26.86%**. And compared with the proprietary model, its capability is close to the upper bound.

These results confirm that our approach effectively integrates time series patterns with textual semantics, substantially boosting both accuracy and generation quality. More evaluations are shown in Appendix G.

## 5.5. Out of Domain Results

We do out-of-domain (OOD) experiments via MT-Bench (Chen et al., 2025b), and we exclude all Weather and Finance data from the training phase, ensuring the setting where the model remains unexposed to these data. Table 2 shows the comparison between PATRA, open-source models, and proprietary models (GPT-4o (OpenAI, 2023), Claude (Anthropic, 2024), and Gemini (Team, 2023)) on these unseen domains. The results demonstrate PATRA's exceptional adaptability. Notably, it exhibits an overwhelming advantage in the Weather domain, achieving state-of-the-art (SOTA) performance across both metrics for both news-driven MCQA and trend prediction. Simultaneously, in the complex Finance domain, it also achieves SOTA performance across all six metrics for both short-term (7-day) and long-term (30-day) trend prediction and numerical forecasting. The details of MTBench are shown in Appendix D.

## 5.6. Ablation Studies

We conduct ablation experiments to assess the contribution of each PATRA component (Table 6). Removing the alignment module causes a substantial drop in reasoning and prescience tasks, highlighting its role in aligning tem-

*Table 2.* Out of domain results about weather news-driven multiple-choice QA (MCQA), weather trend prediction accuracy, finance 3 and 5 stock trend prediction accuracy and finance MACD prediction MSE. The best and second performances are highlighted respectively in **red** with bold and orange with underline. And "-" represents that the model can not generate correct answers.

| Type | Models | Weather | | Finance (7 days) | | | Finance (30 days) | | |
|------|--------|---------|------|------|------|------|------|------|------|
| | | MCQ. ↑ | Pre. ↑ | 3-way ↑ | 5-way ↑ | MSE ↓ | 3-way ↑ | 5-way ↑ | MSE ↓ |
| Proprietary Models | GPT-5.2 | 59.16 | 43.76 | 52.35 | 49.80 | 0.196 | 48.95 | 36.05 | 0.985 |
| | Gemini-3-Pro | 58.74 | 43.68 | 53.11 | 49.79 | 0.209 | 42.20 | 30.27 | 0.999 |
| | GPT-4o | 41.70 | 43.54 | 42.81 | 36.45 | 0.365 | 47.35 | 30.58 | 0.897 |
| | Gemini | 56.96 | 51.76 | 47.30 | 43.40 | 0.384 | 44.90 | 29.70 | 0.975 |
| | Claude | 59.78 | **56.87** | 44.90 | 33.40 | 0.373 | 52.05 | 31.70 | 1.171 |
| | DeepSeek | 46.70 | 25.17 | 45.12 | 35.60 | 0.352 | 48.26 | 29.55 | 1.072 |
| Open-source Models | TimeOmni-1-7B | 41.67 | 51.40 | 40.76 | 34.97 | 0.622 | 37.39 | 20.00 | 1.490 |
| | ITFormer-7B | 57.14 | 36.26 | 40.14 | 33.57 | **0.174** | 51.03 | 31.72 | 0.907 |
| | DeepSeek-R1-7B | – | – | 27.52 | 17.11 | 0.871 | 46.46 | 26.06 | 1.526 |
| | Qwen2.5-7B | 59.00 | 47.77 | 36.71 | 28.04 | 0.458 | 48.38 | 30.28 | 0.887 |
| | ChatTS-7B | 56.21 | 28.79 | 19.71 | 16.09 | 2.047 | 19.38 | 13.46 | 3.342 |
| | **PATRA 7B** | **60.46** | 52.04 | **63.33** | **60.80** | 0.191 | **54.39** | **43.70** | **0.822** |

poral and textual tokens. Using single-pattern alignment also degrades performance, showing that multiple pattern types are necessary for proper understanding. Removing the reasoning-enhanced stage results in the largest drop, emphasizing its importance for building faithful reasoning chains. Our full model consistently achieves the best results, demonstrating the combined effectiveness of pattern-aware alignment and the reasoning-enhanced stage.

*Table 3.* Ablation Studies on PATRA. Due to space limitations, we only report results on the two tasks folowing, while other tasks show similar results in Appendix C.

| Models | Reason. | | Presc. | |
|--------|---------|------|--------|------|
| | Acc. ↑ | Rou. ↑ | Acc. ↑ | Rou. ↑ |
| W/ Single-Pattern Alignment | 35.81 | 26.19 | 37.03 | 24.03 |
| W/O Pattern-Aware Alignment | 28.37 | 16.81 | 30.55 | 16.94 |
| W/O Reasoning-Enhanced Stage | 13.51 | 2.92 | 16.66 | 13.06 |
| PATRA | **44.59** | **27.36** | **52.78** | **27.06** |

## 5.7. Tuned Model vs. Zero-Shot Model

*Table 4.* Performance comparison under Zero-shot and Tuned settings across four tasks.

| Model | Setting | Comp. ↑ | Recog. ↑ | Reason. ↑ | Presc. ↑ |
|-------|---------|---------|----------|-----------|----------|
| DeepSeek-R1-7B | Zero-shot | 12.93 | 12.41 | 18.24 | 14.81 |
| | Tuned | **40.52** | **29.65** | **30.40** | **25.00** |
| TimeMQA-Qwen-7B | Zero-shot | 6.03 | 19.72 | 10.13 | **27.77** |
| | Tuned | **10.34** | **24.41** | **13.51** | 26.85 |
| TimeMQA-Mistral-7B | Zero-shot | 6.90 | 12.41 | 10.81 | 9.26 |
| | Tuned | **18.96** | **26.20** | **12.83** | **28.70** |
| ChatTS-7B | Zero-shot | 25.00 | 28.68 | **38.51** | 12.03 |
| | Tuned | **44.83** | **36.00** | 22.97 | **25.92** |

To validate the effectiveness of our training dataset, we compare the performance of the fine-tuned models against their original zero-shot capabilities on TSQA tasks. As shown in Table 4, the tuned models generally outperform the zero-shot baselines across most metrics, with significant performance gains. These consistent improvements provide strong

empirical evidence that our training data is instrumental in adapting LLMs to the time series domain. By grounding the models in the time series domain, our training data is a vital component in robust time series reasoning.

## 5.8. Case Studies

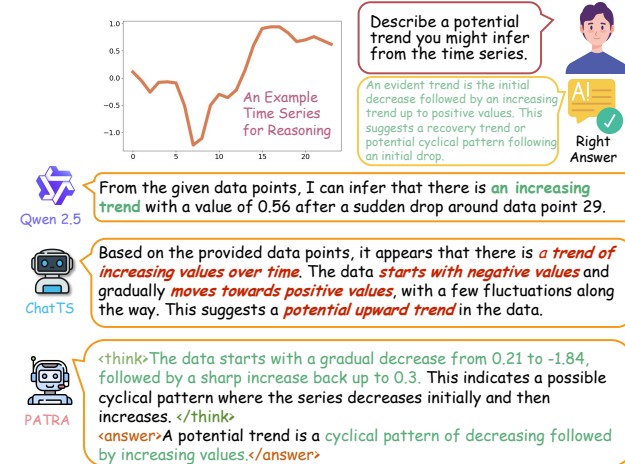

*Figure 4.* Case Study of TS Reasoning Task, comparing the answers of a single-modality LLM (Qwen2.5), a multimodal LLM (ChatTS), and our proposed model (PATRA).

Figure 4 illustrates an example of the TSQA. As observed, a textual LLM (Qwen2.5) fails to describe the series' trend. It incorrectly suggests a simple increasing trend without capturing the series' actual behavior. In contrast, ChatTS, a multimodal LLM with shallow alignment, generates a

response that includes an explicit reasoning chain. However, its reasoning is flawed as it overlooks the cyclical patterns, only focusing on a simple upward trend. As for our proposed model, it delivers a correct answer and integrates deeper reasoning. It identifies the potential cyclical patterns,

*Table 5.* Ablation study on the Reward Balancing Strategy. "Original Reward" refers to training with raw, unscaled rewards, while "Balanced Reward" maps all task rewards to [0, 2].

| Models | Comp. | | Recog. | | Reason. | | Presc. | |
|---|---|---|---|---|---|---|---|---|
| | Acc. ↑ | Rou. ↑ | Acc. ↑ | Rou. ↑ | Acc. ↑ | Rou. ↑ | Acc. ↑ | Rou. ↑ |
| Original Reward | 39.65 | 21.92 | 48.83 | 19.00 | 37.84 | 21.64 | 35.18 | 16.88 |
| **Balanced Reward** | **56.03** | **25.67** | **64.69** | **25.46** | **44.59** | **27.36** | **52.78** | **27.06** |

showcasing enhanced interpretability.

Figure 5 presents a case that highlights a limitation when handling highly volatile, non-stationary sequences. While the Decomposition module separates the signal into trend and seasonal components $X_{ts}^t, X_{ts}^s$, the inherent high variance of this sample introduces ambiguity. The Chain of Thought reveals that the Trend Query $Q_{trend}$ likely assigned excessive attention weights to the specific high-magnitude segment during the cross-modal interaction stage.

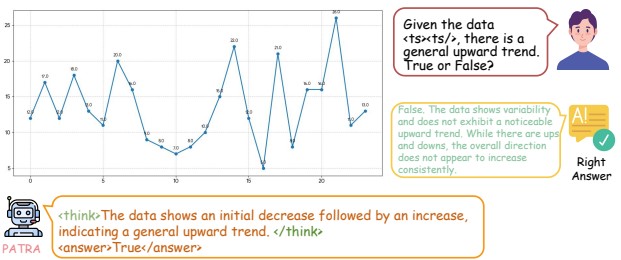

*Figure 5.* Attention bias in Pattern-Aware Alignment.

### 5.9. Reward Comparision.

To validate the effectiveness of the Balanced Reward mechanism, we conduct an ablation study against an "Original Reward" setting, where raw task rewards are directly used without unified scaling. As discussed in the methodology, unbalanced rewards in multi-task reinforcement learning can cause "reward hacking" (Ruder, 2017). Tasks evaluated by Accuracy often produce larger reward magnitudes or easier optimization signals than text generation tasks, leading the model to over-optimize these tasks while neglecting semantic generation abilities. This imbalance weakens the model's understanding of time series semantics and harms overall downstream performance.

The results in Table 5 show that the model trained with Original Rewards performs consistently worse than PATRA across all metrics. This demonstrates that Balanced Reward effectively stabilizes optimization and improves the learning of complex temporal reasoning capabilities.

### 5.10. Latent Decomposition Analysis

To verify whether decomposition in a high-dimensional latent space is meaningful, we conduct a controlled simulation using a synthetic time series with known trend and seasonality. The series is tokenized and projected into the language embedding space, after which our Latent Decomposition module is applied. The decomposed latent features are then projected back to scalar values via PCA for visualization.

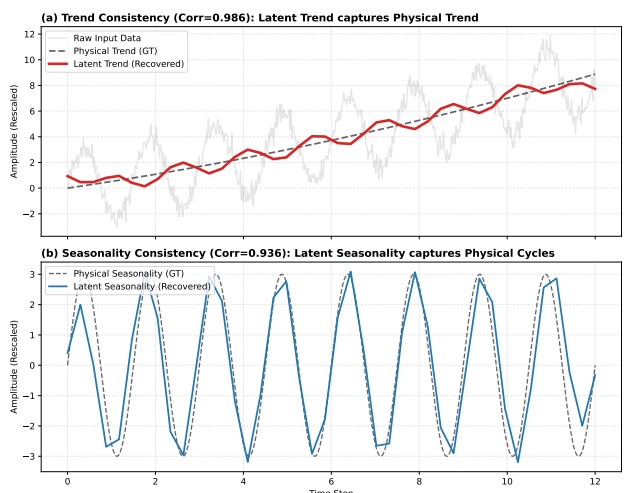

*Figure 6.* The analysis study of Latent Decomposition.

As shown in Figure 6, the extracted latent components closely match the physical signals. The latent trend achieves a Pearson correlation of 0.986 with the ground-truth trend, while the latent seasonality reaches 0.938 correlation with the true periodic pattern. These results provide strong empirical evidence that latent-space decomposition can effectively preserve and disentangle meaningful temporal structures.

## 6. Conclusion

In this paper, we introduce PATRA to advance TSQA from superficial perception to deep reasoning. By disentangling intrinsic temporal dynamics, our Pattern-Aware Alignment ensures precise semantic grounding. Furthermore, our RL-enhanced training with a unified reward mechanism resolves multi-task optimization imbalances, fostering robust Chains of Thought. Experiments confirm that PATRA achieves SOTA performance across benchmarks with exceptional zero-shot generalization.

## Acknowledgements

This work was partially supported by the National Natural Science Foundation of China (62372179, 62406112).

## Impact Statement

This paper presents work whose goal is to advance the field of Machine Learning, specifically in Time Series Question Answering. Since our work involves Large Language Models (LLMs) in Methodology, we want to claim that we use the Qwen2.5 7B model as the backbone for our PA-TRA framework and train it on the publicly available TSQA dataset. Note that though we also used LLMs to polish some grammar during the writing process, all content is original rather than generated, and there are no ethical issues.

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

## A. Dataset Details

### A.1. Dataset Overview.

We conduct experiments on the TSQA dataset (Kong et al., 2025). TSQA introduces a natural language–based reasoning paradigm that bridges classical time series analysis and large language model (LLM) reasoning. It contains approximately 200,000 samples collected from over 12 real-world domains (e.g., energy consumption, stock market, epidemiology, climate). Each sample consists of:

- Raw time series segment.

- Textual context: describing the scenario or asking a question about the series.

- Reference answer: either a textual explanation or a numerical response.

The data is split into training and test sets (rate of 90% and 10%), and the test set is divided into four task categories to enable comprehensive evaluation of temporal reasoning ability:

1. **Comprehension Task**: Focuses on fundamental understanding of the statistical properties of time series. This includes tasks such as identifying maximum or minimum values, summarizing the overall level or range, and describing basic distributional characteristics.

2. **Recognition Task**: Targets pattern identification, anomaly detection, and perception of volatility or seasonality. Models are required to detect recurring trends, sudden spikes or drops, and distinguish normal from abnormal behavior in the series.

3. **Reasoning Task**: Emphasizes higher-order temporal reasoning and logical inference. Tasks involve deducing potential causes, projecting trend continuation, or combining multiple temporal cues to arrive at a conclusion, thus testing the model's ability to perform step-by-step analytical reasoning.

4. **Prescience Task**: Evaluates decision-support capabilities. Models must infer plausible future values, assess risks or opportunities based on historical trends, and provide answers that support planning or action.

This multi-task structure allows us to evaluate not only direct comprehension but also higher-order reasoning and extrapolation abilities of LLM-based models.

## B. Evaluation Prompt

To ensure a fair and reproducible comparison across all models, we adopt a standardized evaluation protocol. Specifically, we design a ***unified prompts*** that are applied identically to every model under evaluation. The prompt for multiple-choice and true/false is shown in following , and we use "You are a helpful assistant" as the prompt for open-ended QA. This eliminates performance variations caused by prompt design differences and isolates the effect of the model architecture and training paradigm itself.

---

**Evaluation Prompt**

You are an assistant for evaluating time series questions.

**Task:**
Answer the question concisely in the required format.

**Tips:**
1. Your thinking steps should be in <think> tags.

2. Always output **exactly one answer**, using the following formats:

- For True/False questions: \boxed{True} or \boxed{False}

- For Multiple Choice questions (options A, B, C, D): \boxed{A}, \boxed{B}, \boxed{C}, or \boxed{D}

---

In addition, we configure the inference process with ***do_sample=False***, disabling sampling-based decoding strategies. This guarantees deterministic outputs for each input, thereby minimizing stochastic variance in model responses. This setup not only improves the reproducibility of results but also enables a fairer comparison of model performance by removing randomness as a confounding factor.

## C. Ablation Studies

To evaluate the contribution of each component in PATRA, we conduct detailed ablation experiments by progressively removing or modifying key modules. Table 6 summarizes the results on reasoning and prescience tasks, reporting both Accuracy and Rouge-L scores.

First, we examine the impact of the alignment module. Removing the pattern-aware alignment (*W/O Pattern-Aware Alignment*) results in a substantial performance drop across all metrics: Accuracy decreases from 44.59% to 28.37% on reasoning tasks and from 52.78% to 30.55% on prescience tasks, while Rouge-L scores drop from 27.36 to

*Table 6.* Ablation Studies on PATRA. Due to space limitations, we only report results on the two tasks folowing, while other tasks show similar results in Appendix C.

| Models | Comp. | | Recog. | | Reason. | | Presc. | |
|---|---|---|---|---|---|---|---|---|
| | Acc. ↑ | Rou. ↑ | Acc. ↑ | Rou. ↑ | Acc. ↑ | Rou. ↑ | Acc. ↑ | Rou. ↑ |
| W/ Single-Pattern Alignment | 44.82 | 24.59 | 58.48 | 23.50 | 35.81 | 26.19 | 37.03 | 24.03 |
| W/O Pattern-Aware Alignment | 40.51 | 15.36 | 47.03 | 15.18 | 28.37 | 16.81 | 30.55 | 16.94 |
| W/O Reasoning-Enhanced Stage | 40.51 | 3.21 | 16.68 | 5.51 | 13.51 | 2.92 | 16.66 | 13.06 |
| **PATRA** | **56.03** | **25.67** | **64.69** | **25.46** | **44.59** | **27.36** | **52.78** | **27.06** |

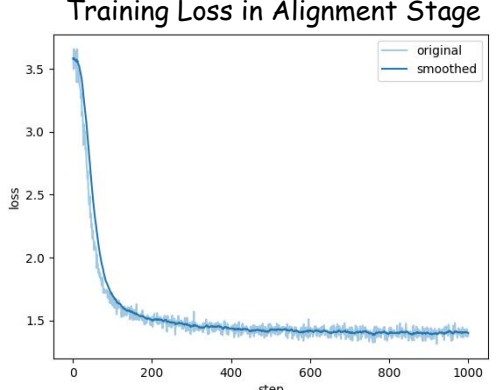
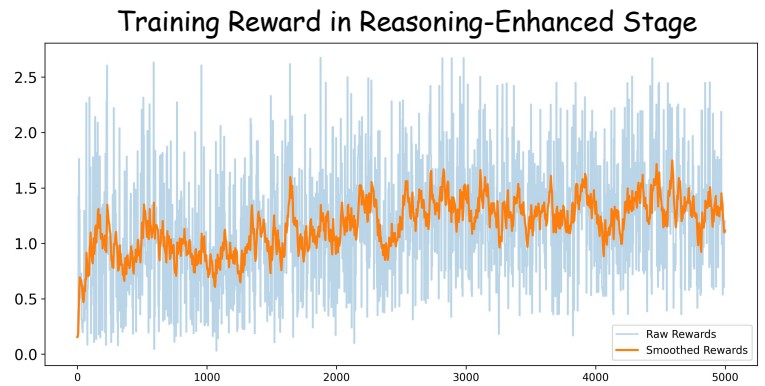

*Figure 7.* The Loss of Alignment stage and the Total Reward Curve in Reasoning-Enhanced Stage.

16.81 and from 27.06 to 16.94, respectively. This clearly demonstrates that pattern-aware alignment is essential for effectively bridging the gap between temporal tokens and textual queries. To further investigate the importance of modeling multiple types of temporal patterns, we replace the pattern-aware alignment with a single-pattern alignment (*W/ Single-Pattern Alignment*). While this variant performs better than removing the alignment entirely, it still underperforms the full model, confirming that relying on a single type of time series pattern is insufficient for comprehensive understanding and reasoning over complex temporal data.

Second, we analyze the effect of the reasoning-enhanced stage. Removing this stage (*W/O Reasoning-Enhanced Stage*) causes the most dramatic degradation, particularly on reasoning tasks, where Accuracy drops to 13.51% and Rouge-L to 2.92. Even prescience tasks are affected, with Accuracy and Rouge-L decreasing to 16.66% and 13.06, respectively. These results indicate that the reasoning-enhanced stage plays a critical role in constructing faithful and coherent reasoning chains, allowing the model to correctly capture dependencies across multiple temporal patterns and generate interpretable chains of thought.

Overall, the ablation study highlights that both components: pattern-aware alignment and reasoning-enhanced stage, contribute significantly to the overall performance of PATRA. The combination of multi-pattern alignment and reasoning-enhanced training enables the model to achieve the best

results across all metrics, verifying the effectiveness of our design for Time Series Question Answering tasks.

## D. MTbench

To evaluate the multimodal reasoning and out-of-distribution (OOD) capabilities of our model, we employ MTBench (Chen et al., 2025b), a large-scale benchmark tailored for assessing LLMs on aligned time series and textual understanding across financial and weather domains.

The dataset comprises 20,000 pairs of stock prices and financial news for the finance subset, and 2,000 pairs of temperature records aligned with synthetic storm reports across 50 U.S. stations for the weather subset. Crucially, the financial data is stratified into consistent and misaligned subsets to rigorously test model robustness against contradictory cross-modal signals, while the weather data synchronizes hourly meteorological readings with event descriptions to accurately reflect evolving climate trends.

While MTBench (Chen et al., 2025b) establishes a comprehensive framework with diverse reasoning-intensive tasks, we focus our evaluation on two core objectives within the Finance and Weather domains. We select weather news-driven multiple-choice QA (MCQA), weather trend prediction accuracy, finance 3, and 5 stock trend prediction accuracy and finance MACD prediction, out-of-domain results about weather news-driven multiple-choice QA (MCQA), weather

trend prediction accuracy, finance 3, and 5 stock trend prediction accuracy and finance MACD prediction MSE. And our model achieves SOTA performance across all metrics.

## E. Training Details

### E.1. Alignment Stage

The Alignment stage plays a crucial role in enabling the model to **understand temporal information**. During this stage, the model is trained on paired temporal data and textual descriptions, allowing it to capture the mapping between time series patterns and natural language. This process equips the model with fundamental temporal comprehension capabilities before performing reinforcement learning or multi-task optimization. The decreasing loss trend shown in Figure 7 confirms that the model is gradually improving its ability to align temporal patterns with textual semantics, laying a solid foundation for subsequent reasoning and decision-making stages.

### E.2. Reasoning-Enhanced Stage

#### E.2.1. REWARD COMPOSITION.

The total reward is composed of two parts: *A task reward* that reflects the correctness and reasoning quality of the model output, which is scaled to fall within $[0, 2]$. Rewards in $[0, 1]$ led to weaker policy gradients and slower convergence, whereas $[0, 2]$ produced sufficiently strong learning signals without destabilizing updates. *A format reward* that encourages well-structured responses (e.g., correct use of <think> and <answer> tags), which ranges from $[0, 0.75]$. As a result, the overall reward may exceed 2 for well-formed and fully correct answers.

#### E.2.2. TRAINING STABILITY.

Figure 7 shows the training reward curve. While the raw rewards (**blue**) exhibit high variance due to the heterogeneous nature of tasks, the exponentially smoothed rewards (**orange**, EMA with $\alpha = 0.04$) remain stable, indicating stable optimization. The slight reward values above 2 are expected, as they represent cases where the model not only answers correctly but also produces perfectly formatted outputs. This confirms that our reward scaling and GRPO optimization yield a stable training process without mode collapse, even in the presence of multi-type task heterogeneous rewards.

## F. Other Related Work

Time Series Reasoning has recently attracted growing attention due to its potential to enable intelligent understanding and decision-making from temporal data. Unlike conventional forecasting tasks that mainly focus on predicting future values, Time Series Reasoning aims to answer complex questions based on time-dependent signals, requiring both temporal perception and logical reasoning abilities. Such capabilities are valuable in many real-world applications, including economics (Wu et al., 2024; Wang et al., 2025a), transportation (Qiu et al., 2026b; Wang et al., 2026b; Wu et al., 2026a; Ma et al., 2014; Yang et al., 2023), healthcare (Wu et al., 2025a; Wang et al., 2026c; Cheng et al., 2026a), energy systems (Wang et al., 2026a; Liu et al., 2026a; Cheng et al., 2026c; Tian et al., 2026; tian et al., 2025), and AIOps (Yu et al., 2025a).

Existing Time Series Reasoning and forecasting methods still face significant challenges. Many approaches struggle to effectively capture important temporal patterns such as trends, seasonality, cross-variable dependencies, and irregular temporal dynamics, especially under missing values and distribution shifts (Qiu et al., 2025d;b; 2026a; Liu et al., 2026b;a; Wu et al., 2025c;b; Li et al., 2026c; Wu et al., 2025d; Yu et al., 2025b). To address these issues, recent studies have explored frequency-domain modeling (Wang et al., 2025a; Lu et al., 2026), graph and hypergraph neural architectures (Chen et al., 2023; Shang et al., 2024; Shang & Chen, 2024; Shang et al., 2026), probabilistic modeling (Wu et al., 2025c; Cheng et al., 2026c), multimodal semantic alignment (Hu et al., 2026a), and foundation-model-based paradigms (Cheng et al., 2026b; Wang et al., 2026a; 2025d;c). Meanwhile, comprehensive benchmarks and evaluation frameworks have also been proposed to standardize forecasting and anomaly detection research (Qiu et al., 2024a; 2025a; Shentu et al., 2025; Li et al., 2025). Recent studies further extend time series understanding toward anomaly prediction and detection tasks through multi-scale modeling, adaptive period learning, and role-aware channel interactions (Hu et al., 2024; 2026c;b).

Beyond traditional time series analysis, researchers have increasingly leveraged the representation learning capability of deep neural networks (DNNs) and multimodal learning frameworks (Fang et al., 2023b; 2022; 2026a; 2023a; Li et al., 2026b;a; Zhang et al., 2024; 2025a;b;c; Ni et al., 2026; Chen et al., 2025c; Qi et al., 2025; Wang et al., 2026d; Ma et al., 2024; 2025b; 2022; 2026; 2025a; Yu et al., 2025d;c;e; Lu et al., 2024a; 2023; 2024b; Wu et al., 2026b). These advances have demonstrated the strong potential of large-scale representation learning for complex reasoning and cross-modal understanding. Inspired by these successes, recent studies have begun exploring the integration of large language models with time series modeling (Shang et al., 2026; Fang et al., 2026b;c), opening a promising direction toward more general and intelligent Time Series Reasoning systems.

*Table 7.* Three Common Time Series Tasks' Evaluations. We use the Accuracy metric for Anomaly Detection and Classification tasks and the MSE metric for the Forecasting task.

| Type | Models | Forecasting MSE ↓ | Anomaly Detection Acc. ↑ | Classification Acc. ↑ |
|---|---|---|---|---|
| Proprietary Models | Doubao | - | 0.52 | 0.44 |
| | GPT-4o | 1.79 | 0.64 | 0.32 |
| | backbones | | *TimeMQA* | |
| Open-source Models | Llama-3-8B | 2.01 | 0.54 | 0.24 |
| | Qwen-2.5-7B | 1.82 | 0.68 | 0.52 |
| | Mistral-7B | 1.35 | 0.58 | 0.44 |
| | PATRA-7B | **1.08** | **0.74** | **0.60** |

# G. Common Time Series Tasks Evaluation

To verify the generalization capabilities of PATRA in common time series tasks, we evaluate the model on three fundamental downstream tasks following TimeMQA's setting (Kong et al., 2025): Forecasting, Anomaly Detection, and Classification. The detailed evaluation results are presented in Table 7.

In the experiments, we utilize Mean Squared Error as the metric for the forecasting task and Accuracy for both anomaly detection and classification tasks. We compare PATRA-7B against proprietary models (Doubao (Chen et al., 2025a), GPT-4o (OpenAI, 2023)) and TimeMQA (Kong et al., 2025) series models including Llama-3-8B (Grattafiori et al., 2024), Qwen-2.5-7B (Yang et al., 2025), and Mistral-7B (Jiang et al., 2023) backbones.

The results demonstrate that PATRA-7B achieves the best performance across all three tasks, significantly outperforming the baselines: PATRA achieves the lowest MSE of *1.08*, surpassing the best open-source baseline Mistral-7B (1.35) and the proprietary GPT-4o (1.79); Anomaly Detection: Our model attains an accuracy of *0.74*, outperforming both Qwen-2.5-7B (0.68) and GPT-4o (0.64); Classification: PATRA shows a substantial advantage with an accuracy of *0.60*, significantly higher than the runner-up Qwen-2.5-7B (0.52) and GPT-4o (0.32).

### G.1. Forecasting Evaluation Prompt Example

**Forecasting Evaluation Prompt Example**

**System Prompt:**

You are a time series forecasting assistant.
You **MUST** generate a prediction sequence for every request, regardless of uncertainty or data quality.
Do not provide explanations, apologies, or text like **I cannot predict**.
Your output must contain **ONLY** the numerical sequence in the format: [value1, value2, ...].

**Query:**

The London Smart Meters Dataset contains half-hourly energy consumption readings in kWh from 5,560 households in London, collected between November 2011 and February 2014, and categorized into 112 blocks. The frequency is one hour. The input Time Series are <ts><ts/>. Please predict the next 31 time series points given the information above.

**Time series:**

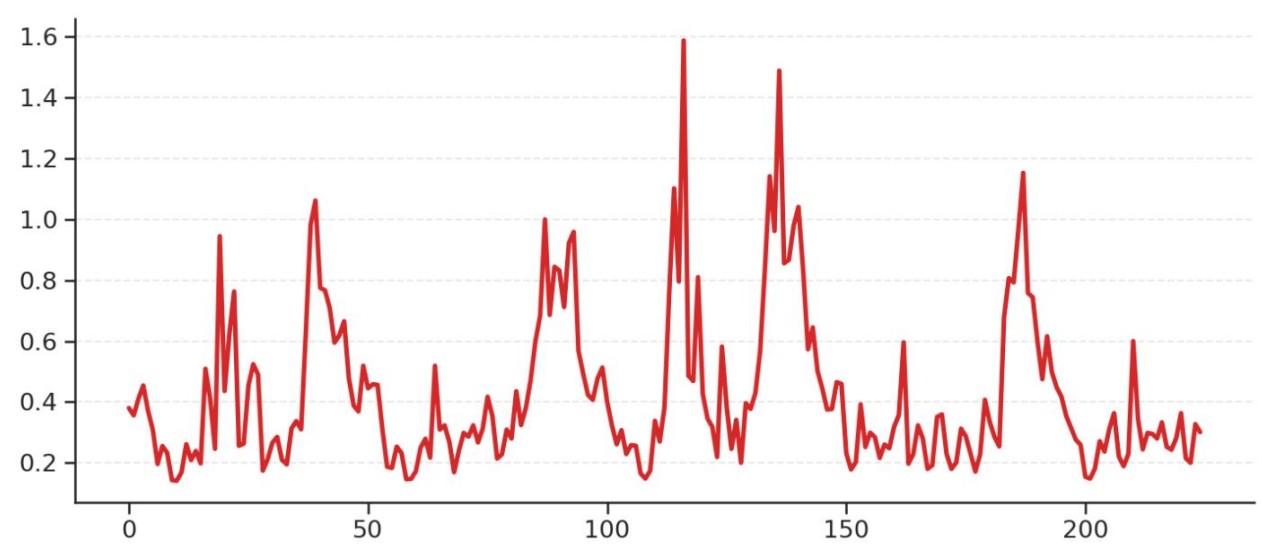

### G.2. Anomaly Detection Evaluation Prompt Example

**Anomaly Detection Evaluation Prompt Example**

**System Prompt:**

You are a time series anomaly detection assistant.
You **MUST** generate an answer for every request, regardless of uncertainty or data quality.
Do not provide explanations, apologies, or text like **I cannot predict**.
Your output must contain **ONLY** one word chosen from ['anomaly', 'normal'].

**Query:**

The following data represents ECG (Electrocardiogram) signals, which record the electrical activity of the heart during each heartbeat and serve as an important tool for diagnosing heart diseases. Using a signal sampled at a frequency of 500 Hz, we can determine whether abnormalities are present in the human body. The input Time Series is <ts><ts>. Please determine whether there are anomalies in this time series, given the information above.

**Time series:**

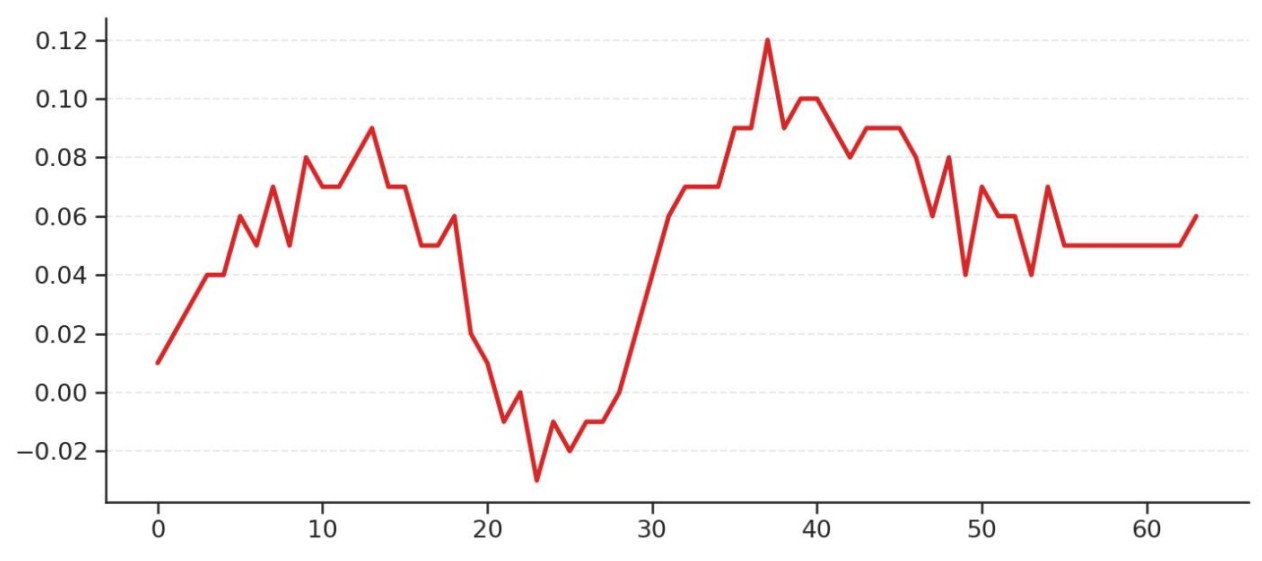

### G.3. Classification Evaluation Prompt Example

**Classification Evaluation Prompt Example**

**System Prompt:**

You are a time series classification assistant.
You **MUST** generate an answer for every request, regardless of uncertainty or data quality.
Do not provide explanations, apologies, or text like **I cannot predict**.
Your output must contain **ONLY** one or two words chosen from ['no freeze', 'freeze'], or ['walking', 'jogging', 'upstairs', 'downstairs', 'sitting', 'standing'].

**Query:**

The following data provides accelerometer data for activity recognition research. The dataset has a sampling rate of 20Hz and records accelerometer data for six activity states: walking, jogging, sitting, standing, upstairs, and downstairs. Each sample includes acceleration values for the X, Y, and Z axes, ranging from -20 to 20, where 10 represents 1g, and 0 indicates no acceleration. The recorded acceleration includes gravitational acceleration, so when the phone is stationary on a flat surface, the vertical axis registers approximately. We provide 10 timestamps of accelerometer data, with each timestamp containing X, Y, and Z values, for a total of 30 values. The recorded Time Series is <ts><ts>. Please judge whether this data corresponds to 'Walking' or 'Jogging' or 'Upstairs', 'Downstairs' or 'Sitting' or 'Standing', given the information above.

**Time series:**

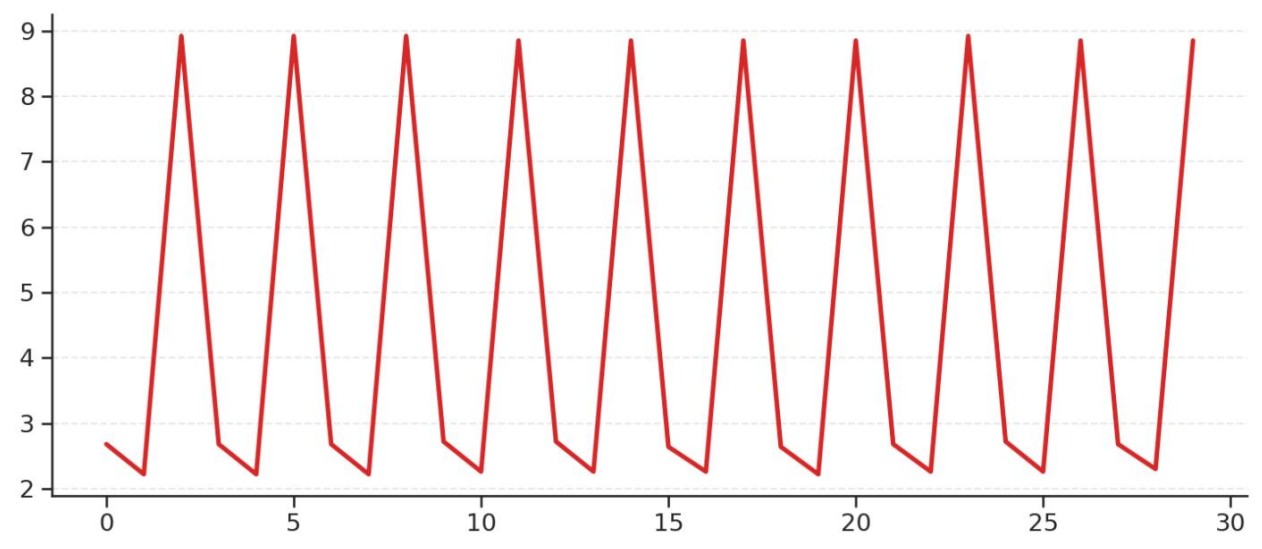

