# OpenReview forum: "PATRA: Pattern-Aware Alignment and Balanced Reasoning for Time Series Question Answering"
_ICML.cc/2026/Conference — ICML 2026 regular_

### Official Review · Reviewer_5hyg · 2026-02-18

**Soundness:** 3
**Presentation:** 4
**Significance:** 3
**Originality:** 2
**Overall Recommendation:** 5
**Confidence:** 3

**Summary:**

PATRA is a framework for time series question answering (TSQA) that addresses two main challenges: cross-modal alignment and stable inference across multiple tasks. The model first performs an explicit decomposition of time series representations, separating them into structural components such as trend and seasonality. Based on this decomposition, it introduces a pattern-aware alignment mechanism that connects temporal structures in the data with the semantics of textual queries, allowing language reasoning to better reflect the underlying behavior of real-world time series. To further improve training, PATRA adopts a task-aware balanced reward strategy that mitigates optimization bias among heterogeneous tasks in the reinforcement learning setting, leading to more consistent performance across tasks of varying difficulty. Experiments on the TSQA benchmark show that PATRA surpasses several strong baselines on multiple evaluation metrics and maintains robust performance in out-of-domain generalization, ablation studies, and standard time series analysis tasks.

**Compliance With Llm Reviewing Policy:**

Affirmed.

**Final Justification:**

Recommended to accept

**Key Questions For Authors:**

1.The trend/seasonality decomposition currently seems fairly simple, essentially built on average pooling plus residual separation. I am not fully convinced this is the most appropriate design choice. It would strengthen the paper if the authors could compare it against a few stronger alternatives, such as STL-style decomposition, frequency-based methods, or learnable temporal filters. I would also like to see what happens if this explicit decomposition step is removed entirely. If the gap is small, then the necessity of this part of the framework becomes less clear.

2.The paper claims that the pattern-aware alignment leads to better cross-modal grounding, but at the moment this is mostly supported by end-task performance. I think the paper would be more convincing if the authors could provide some direct evidence for this claim. For example, attention visualizations, alignment diagnostics, or more targeted ablations on different branches of the alignment module would help show whether the improvement באמת comes from the alignment design itself, rather than from overall training or model capacity.

3.For the multi-task RL part, I would like to better understand how important the reward normalization actually is. Have the authors tried a version without this normalization, or compared it with other simple scaling strategies? Right now it is a bit hard to tell whether the gains come from a principled balancing effect across tasks, or simply from reward shaping in a broader sense.

4.It would also be useful to report some measure of result stability for the main experiments. Standard deviations, confidence intervals, or significance tests would make it easier to judge whether the reported improvements are consistently reproducible across runs, especially since RL-based training can sometimes be quite noisy.

**Limitations:**

No. The paper provides only a limited discussion of its technical constraints and broader implications. It would be beneficial to briefly elaborate on the conditions under which the method may be less effective, such as handling highly irregular or complex time series, and to acknowledge potential uncertainties when applied in practical decision-making settings. Adding a discussion of limitations will help improve the completeness and rigor of the paper.

**Strengths And Weaknesses:**

Strengths

1.The paper tackles a worthwhile and fairly timely problem, namely extending large language models to the setting of time series question answering. This remains a relatively less explored area in multimodal reasoning, and the paper does a good job explaining why the problem matters in practice, especially in terms of shallow cross-modal alignment and the difficulty of balancing different task types during training.

2..The pattern-aware alignment mechanism introduces a structured approach to integrating time series and textual representations by explicitly decomposing temporal data into trend/seasonality components before alignment. This reflects an awareness of the intrinsic properties of time series data and goes beyond simple embedding concatenation.

3.The task-aware balanced reward design is also practically motivated. Since the tasks in TSQA can differ substantially in difficulty and format, introducing reward normalization and staged supervision seems like a reasonable attempt to make multi-task RL training more stable.

4.The experiments are fairly broad and cover several relevant settings. In addition to the main benchmark results, the paper includes out-of-domain evaluation and ablation studies, and the improvements over competitive baselines appear generally consistent rather than isolated to a single case.

5.The paper is also easy to follow. The overall framework, the training pipeline, and the experimental protocol are presented in a fairly clear order, and the level of detail is mostly sufficient for readers to understand how the method is implemented.

Weaknesses

1.Some of the core components still feel technically lightweight. In particular, the decomposition of time series patterns seems to rely on relatively simple operations such as average pooling and residual separation. This makes the design intuitive, but also somewhat limited, and I am not sure it is strong enough to fully support the paper’s broader claims about structured temporal modeling.

2.The reinforcement learning part is useful from an engineering perspective, but the novelty appears more moderate than the paper sometimes suggests. Most of the contribution here seems to come from reward shaping and balancing strategies built on top of an existing optimization framework, rather than from a genuinely new RL algorithm.

3.The empirical improvements are encouraging, but the paper still leaves an important “why” question insufficiently answered. In particular, there is not much direct evidence showing how or why the pattern-aware alignment improves reasoning. More interpretability analysis, qualitative examples, or visualization of the alignment behavior would make this part of the argument much more convincing.

4.In terms of originality, the work reads more as a thoughtful combination of existing ideas than as a fundamentally new methodological step. That does not make the contribution unimportant, but it does place it more on the side of solid system design and engineering refinement than deeper algorithmic innovation.

5.The broader impact of the work may also be somewhat narrow. The method seems well suited to TSQA and perhaps to related forms of structured multimodal reasoning, but its implications for more general machine learning methodology or theory are less obvious.

6.The presentation could be polished further. Figure 1 is hard to read because some of the text is too small, and the current table styling relies rather heavily on color highlighting, which can be distracting. A cleaner visual design would likely improve readability and help the reader focus more easily on the main results.

---

> ### Author Rebuttal · Authors · 2026-03-30
>
> We sincerely thank the reviewer for the constructive and insightful feedback. We appreciate the thoughtful comments, which help clarify the aspects of our design and evaluation. Below, we address each concern in detail.
>
> ### Q1. The decomposition method seems simple.
>
> Response:
> We agree more sophisticated decomposition methods are powerful in classical time series analysis. However, our goal is not to design a stronger decomposition module, but to ensure stable and effective cross-modal alignment with LLMs.
>
> 1. We first verify the necessity of our design through ablation (Table 5): removing the decomposition (W/O Pattern-Aware Alignment) leads to ~20% performance drop, showing that this component is essential rather than optional.
> 2. We also explored more complex alternatives, but observed limitations in the cross-modal setting:
>     - Frequency-based methods introduce representations that are not naturally aligned with the LLM embedding space, making integration less stable;
>     - Learnable filters, while more expressive, tend to overfit the specific patterns and may harm generalization.
>
> ### Q2. The RL contribution is not that novel.
>
> Response:
> We agree that our contribution is not a new RL optimizer, but a reward design for TSQA. In TSQA, tasks vary in difficulty and reward structure, and standard RL tends to over-optimize easier tasks, leading to imbalanced learning. Our balanced reward normalization maps heterogeneous rewards into a unified space, rebalancing optimization across tasks (more details are in reviewer a5JJ’s Q2).
>
> As shown in Appendix E.2.2 (Table 6), removing this mechanism leads to ~16% drop in accuracy and ~6% drop in Rouge, indicating its effect goes beyond simple scaling.
>
> ### Q3. Is there direct evidence that improvements come from the alignment module?
>
> Response:
> Thank you for this suggestion. We provide targeted ablations in Table 5 to isolate the effect of the alignment module.
>
> In “W/O Pattern-Aware Alignment”, we keep the same LLM backbone, feature extractor, and RL training pipeline, and only remove the alignment mechanism. This results in ~20% performance drop, indicating that the gains primarily come from alignment rather than model capacity or training procedure.
>
> Additionally, using a explicilty single trend pattern (W/ Single-Pattern Alignment) leads to ~10% drop, suggesting that different patterns provide complementary information. These results collectively validate the effectiveness of pattern-aware alignment.
>
> ### Q4. The work appears more like integration than fundamental novelty.
>
> Response:
> We acknowledge that our contribution is primarily architectural, rather than introducing a new standalone RL algorithm. However, it is not a simple combination of existing components, but is driven by two key challenges in TSQA: cross-modal alignment and task heterogeneity.
>
> - We introduce pattern-level alignment in latent space for structured interaction between time series patterns and text (more details are in reviewer a5JJ’s Q2(1));
> - We design a reward mechanism to address optimization imbalance across tasks (more details are in reviewer a5JJ’s Q2(2)).
>
> ### Q5. The broader impact may be limited beyond TSQA.
>
> Response:
> We agree that our experiments focus on TSQA. That said, the underlying idea, aligning structured continuous signals with LLM semantic representations via pattern-aware decomposition, may generalize to other “structured signal + language” settings.
>
> For example, it could apply to domains involving structured signals where alignment with language is required. We will clarify this potential more carefully in the final version.
>
> ### Q6. The presentation could be improved.
>
> Response:
> Thank you for this suggestion. We will increase font sizes and improve layout in Figure 1, simplify table formatting, and reduce excessive color highlighting. These changes will improve readability in the final version.
>
> ### Q7. Stability metrics would strengthen the results.
>
> Response:
> We agree reporting stability is important, especially for RL-based training. Due to computational constraints during the rebuttal period, we are unable to run multiple seeds at this stage. Nevertheless, we provide two supporting observations:
>
> - Training stability: as shown in Figure 5, while raw rewards exhibit expected variance, the EMA curve converges smoothly without instability or collapse;
> - Performance margins: the observed improvements (10–20%) are substantially larger than typical RL variance, suggesting robustness.
>
> ### Q8. Discussion on limitations.
>
> Response:
> We agree that discussing limitations is important for completeness and rigor. We will include these points in the final version:
>
> 1. The method relies on regular time series patterns; for highly irregular or structureless series, decomposition may be less informative.
> 2. Performance may degrade under significant distribution shifts or unseen temporal dynamics.

---

> > ### Author Rebuttal · Reviewer_5hyg · 2026-04-01
> >
> > Thank you for the clarifications. My concerns have been sufficiently addressed, and I will increase my score accordingly.

---

> > > ### Author Response · Authors · 2026-04-01
> > >
> > > We would like to thank Reviewer 5hyg for the detailed and valuable review, which has greatly helped us improve the clarity and quality of the paper.
> > >
> > > We truly appreciate your constructive feedback and are glad that our clarifications have addressed your concerns. We will incorporate these suggestions into the final version.

---

### Official Review · Reviewer_gA2i · 2026-03-10

**Soundness:** 3
**Presentation:** 4
**Significance:** 3
**Originality:** 3
**Overall Recommendation:** 3
**Confidence:** 3

**Summary:**

This paper studies Time Series Question Answering (TSQA) and aims to improve both cross-modal understanding and multi-step reasoning for temporal data. The paper mainly focuses on two main issues, one is how to better connect time series with natural language, and another is how to train a unified model for different tasks. The authors propose PATRA, a framework that combines a pattern-aware alignment module with a reasoning-oriented training strategy. The alignment module breaks time series into components such as trend and seasonality, then matches them with the input query. Further, the model uses reinforcement learning with a task-aware balanced reward to support reasoning, and reduce training imbalance across tasks. Experiments on several TSQA benchmarks show that PATRA performs better than a range of open-source baselines, and achieves strong results on reasoning and forecasting tasks.

**Compliance With Llm Reviewing Policy:**

Affirmed.

**Final Justification:**

Although I still have some concerns about the core modeling ideas, the authors’ rebuttal addressed part of my concerns, so I have decided to lower my confidence score.

**Key Questions For Authors:**

1. What is actually novelty in the pattern-aware alignment module, in contrast to a standard decomposition with attention design?

2. Can the authors show more clearly that the task-aware balanced reward does more than simple reward rescaling or normalization?

3. How does the method compare with stronger and more relevant baselines, especially recent TS reasoning or LLM-based time-series models?

4. Are there any results that can verify how much of the improvement comes from the alignment module itself, and how much comes from the reasoning-enhanced training stage?

**Limitations:**

Yes

**Strengths And Weaknesses:**

**Strengths**

1. The targeted Time Series Question Answering is an important and timely problem, especially because it sits at the intersection of temporal pattern modeling and language reasoning. The motivation is well defined, because the paper does a reasonable job explaining why existing LLM-based methods still struggle in such setting.

2. The method is organized in a good natural way. The pattern-aware alignment stage and the reasoning-enhanced training stage fit together logically. It makes the overall story easy to follow.

3. Experimental part is easy to follow. The reported gains over the included baselines appear consistent across several TSQA task types. And the additional out-of-domain tests in weather and finance make the empirical section more convincing.

4. There are some useful experiment analysis. Performance drops noticeably when the alignment module or the reasoning-enhanced stage is removed, which suggests that these components are contributing something meaningful rather than being added only for presentation.

**Weaknesses**

1. I think the methodological novelty is not very convincing. The so-called pattern-aware alignment still looks quite close to a standard decomposition, which is mainly supported by attention module, and it is not very clear what is truly new beyond combining these pieces.

2. I am not fully convinced by the RL-based task-aware balanced reward. As written, it seems closer to a reward rescaling or normalization strategy, and the paper does not provide enough evidence that it really solves cross-task optimization imbalance.

3. The experimental comparison feels somewhat limited. The baseline set is relatively narrow, and some relevant TS reasoning systems mentioned in the paper, such as ITFormer and TimeOmni-1, are not included in the main results. And even for the llm-based methods, there are few advanced baselines.

4. The ablation study is still not enough to clearly explain where the gains come from. I would have liked to see stronger comparisons against simpler alternatives, such as stronger SFT-only training or a more standard RL objective without the proposed balancing design.

---

> ### Author Rebuttal · Authors · 2026-03-30
>
> We sincerely thank the reviewer for the critical and insightful feedback. Based on the raised concerns and questions, we organize our responses around the following key questions. Below, we address each point in detail.
>
> ### **Q1. Is the method simply combining standard decomposition with attention?**
>
> Response:
> The key novelty of PATRA lies **not in the decomposition itself, but in performing decomposition in the LLM latent space** for **pattern-level cross-modal alignment**.
>
> While simple in form, our design differs fundamentally from conventional pipelines that decompose in the **raw space** followed by independent embedding, or rely on implicit attention without explicit pattern disentanglement. In contrast, PATRA keeps patterns in a **unified latent semantic space**, avoiding representation mismatch and enabling deep alignment between time series patterns and language.
>
> This enables **pattern-level** interaction with language tokens, shifting from sequence-level modeling to fine-grained cross-modal alignment, where it can be explicitly aligned with textual semantics.
>
> Appendix G shows strong consistency with real patterns (Pearson = **0.986**), and Table 5 shows significant degradation when removing this design.
>
> ### **Q2. Is the reward mechanism merely normalization?**
>
> Response:
> Thank you for this important question. Our key point is that the mechanism **rebalances optimization across heterogeneous tasks**, rather than performing simple rescaling.
>
> - In TSQA, tasks differ significantly in both reward type and difficulty (e.g., classification vs. generative reasoning). Under standard RL, models tend to over-optimize easier tasks, leading to biased learning and insufficient reasoning capability.
> - Our balanced reward normalization maps heterogeneous rewards into a **unified optimization space**, thereby **modifying their relative gradient contributions and changing the optimization trajectory**. This **encourages the model to allocate sufficient capacity to harder reasoning tasks**. We validate this with ablations:
>     - Standard RL (Table 6): **~16%** drop
>     - SFT only (Table 5): **~30%** drop
>
> If the method were merely rescaling, such differences would not be expected. Instead, the results indicate that it fundamentally alters training dynamics and improves reasoning performance.
>
> ### **Q3. How does the method compare with stronger baselines?**
>
> Response:
> Thank you for highlighting this. We agree that comprehensive comparisons are essential.
>
> During rebuttal, we expanded baselines to include recent time-series models (**TimeOmni-1**, **ITFormer**) and strong proprietary LLMs (**GPT-5.2**, **Gemini 3 Pro**) on the four TSQA tasks and **OOD evaluations** (*additional Table 1 and Table 2*, [https://anonymous.4open.science/r/PATRA-38E7/Rebuttal.md)](https://anonymous.4open.science/r/PATRA-38E7/Rebuttal.md).
>
> Overall:
>
> - **PATRA** consistently outperforms **TimeOmni-1** and **ITFormer** across both in-domain and OOD settings;
> - PATRA remains comparable to strong proprietary models such as GPT-5.2 and Gemini-3-Pro, despite using a much smaller model (**7B vs. significantly larger-scale proprietary models**).
> - More importantly, PATRA consistently achieves substantially higher generation quality across all tasks, indicating better alignment with target answers.
> - In addition, **OOD** results further show that PATRA maintains superior performance in **OOD scenarios**, demonstrating stronger generalization ability.
>
> These results indicate that the improvements are not due to weak baselines or overfitting, but stem from effective pattern-aware modeling and training, which generalize well to unseen domains.
>
> ### **Q4. How do alignment, RL, and reward design contribute to performance gains?**
>
> Response:
> Thank you for this important question. We provide a quantitative disentanglement of the contributions from alignment and reasoning-enhanced training using ablations in Table 5 and Table 6.
>
> - Removing the **alignment module** (Table 5) leads to ~**20%** performance drop, indicating its critical role in providing structured temporal representations for cross-modal reasoning;
> - Removing the **reasoning-enhanced training stage** (**SFT-only training**, ****Table 5) leads to ~**30%** drop, showing that reasoning-enhanced training is essential for improving complex reasoning capability;
> - Using standard RL without **balanced reward** (Table 6) results in ~**16%** drop, demonstrating that the performance gain is not only from RL itself, but also from the proposed reward design.
>
> These results show that alignment provides the representation foundation; RL enhances reasoning ability, and task-aware reward further stabilizes optimization across tasks.

---

> > ### Author Rebuttal · Reviewer_gA2i · 2026-04-03
> >
> > Thank you for the detailed rebuttal. I appreciate the added clarification, but my overall view has not changed much.
> >
> > 1. On the alignment side, you argue that the novelty lies in performing decomposition in the latent space and aligning it with language. However, from the implementation, the method still looks largely like a combination of time-series embedding, simple decomposition via average pooling and residual subtraction, learnable alignment tokens, and attention-based fusion. I can totally understand your idea, but I still do not see clear evidence that doing this in latent space, by itself, constitutes a strong methodological novelty, or that it enables something clearly beyond simpler raw-space decomposition or more direct alignment strategies. Relatedly, I remain unconvinced by the claim around “latent reasoning.” The synthetic analysis in Appendix G shows that the latent trend correlates with the physical trend, which just supports feasibility, but not necessarily real reason.
> >
> > 2. I also still find the RL contribution somewhat overstated. My original concern was that the task-aware balanced reward looks closer to reward rescaling / normalization than to a genuinely new optimization mechanism. The rebuttal shows that this design is useful, but improved performance alone does not establish stronger methodological contribution, especially since reward scaling can naturally have a large effect in heterogeneous multi-task RL.
> >
> > 3. More broadly, I think the paper still uses “reasoning” in a fairly loose way. The current evidence seems more consistent with improved cross-modal representation and output-level optimization than with a clearly verified deeper temporal reasoning process.
> >
> > 4. Although the method is presented as backbone-free, all experiments are conducted on Qwen2.5-7B, so there is still limited evidence for generality across different LLM backbones or scales.
> >
> > Overall speaking, I appreciate you reply, and the rebuttal improves the empirical story, but it does not sufficiently resolve my main concern about originality. I therefore would like to maintain my original score. But as a bonus for partially resolving, I'll reduce my confidence from 5 to 4.
> >
> > --
> >
> > Thank you for the additional response. I appreciate the effort you put into the rebuttal. Your clarification helped address some of my uncertainty, and I will further lower my confidence while raising my assessment of the paper’s soundness and presentation.

---

> > > ### Author Response · Authors · 2026-04-04
> > >
> > > We sincerely thank the reviewer for the thoughtful follow-up and for clarifying the remaining concerns. Based on the feedback, we summarize the key issues as follows and provide clarifications.
> > >
> > > ### Q1. Novelty of the alignment and clarifying the “latent reasoning”
> > >
> > > 1. Novelty of the alignment
> > >
> > > The alignment novelty of PATRA does not lie in inventing completely new mathematical operators, but rather in proposing a paradigm shift for cross-modal alignment in TSQA.
> > >
> > > Existing TSQA approaches either treat time series strictly as text tokens or rely on shallow alignment (directly concatenating tokenized time-series patches with text embeddings). This prevailing paradigm limits interaction between intrinsic time series patterns and language semantics. To address this, we propose the **deep alignment** paradigm driven by **Latent Space Decomposition**. We organize time series into distinct patterns in the latent space and align them with language at a finer granularity. This enables pattern-level interactions between time series patterns and language tokens within a unified semantic space.
> > >
> > > 2. Clarify the latent reasoning
> > >
> > > Regarding the concern about "latent reasoning," we would like to politely clarify that **we do not propose or use the term "latent reasoning" in our paper**. We claim that organizing and aligning time series in the latent space fundamentally enhances the LLM's overall time series **reasoning capability.**
> > >
> > > In Table 5 of the paper, removing the Pattern-Aware Alignment causes a significant absolute drop of **16.22%** (from **44.59% to 28.37%**) in Reasoning accuracy. This provides strong empirical evidence that our proposed alignment, driven by Latent Space Decomposition, effectively enhances the model's time series reasoning performance.
> > >
> > > ### Q2.  Reward Design
> > >
> > > We agree that the implementation involves adjusting reward signals across tasks. However, the goal is to address the inherent imbalance between tasks of varying difficulty and reward characteristics in TSQA.
> > >
> > > In practice, different task types (e.g., classification vs. generative reasoning) provide rewards with different properties. Without adjustment, models **tend to rely more on the easier tasks, which can lead to insufficient learning of more complex reasoning tasks**. Our design introduces a task-aware balanced reward to **ensure that all tasks are fairly represented during training**. I**ts primary role is to rebalance the learning process across heterogeneous tasks**.
> > >
> > > ### Q3. Use of the term “reasoning”
> > >
> > > We sincerely thank the reviewer for the careful perspective on terminology. We agree that the primary contributions of PATRA lie in cross-modal alignment and RL-based training design. However, we would like to clarify that, in the specific context of TSQA, such foundational enhancements are precisely what enable models to exhibit reasoning capabilities over time series.
> > >
> > > As discussed in recent works that establish the TSQA setting (e.g., TimeOmni-1, ITFormer, ChatTS), “reasoning” in time series typically refers to the ability of a model to identify underlying time series temporal dynamics and derive logically consistent conclusions based on them. In PATRA, our latent-space decomposition organizes time series into structured, fine-grained components within a unified semantic space with text, which facilitates more explicit interaction between time series patterns and language semantics.
> > >
> > > For example, as shown in Figure 4, PATRA generates intermediate reasoning traces within the `<think>` tags, where the model connects specific time series dynamics (e.g., a gradual decrease followed by a sharp increase) to the final semantic conclusion (e.g., periodic patterns).
> > >
> > > ### Q4. Backbone dependency
> > >
> > > We sincerely thank the reviewer for pointing this out. We agree that evaluating across multiple backbones will further strengthen the claim of generalization.
> > >
> > > First, we would like to clarify the focus on Qwen2.5-7B in our experiments. The latest and most competitive TSQA models, such as **TimeOmni-1, ITFormer, and ChatTS**, all use a similar backbone (Qwen2.5-7B or larger-scale variants). By conducting our experiments on Qwen2.5-7B, we ensure a strict and fair comparison with these state-of-the-art baseline models. Our results show that the improvements stem directly from the proposed alignment mechanism and reward design, rather than from using a superior backbone.
> > >
> > > Second, due to the time constraints during the rebuttal phase, we were unable to complete comprehensive cross-backbone training and evaluation. However, we want to emphasize that the proposed alignment and reward design is inherently **backbone-agnostic**. We are committed to including additional experiments on other standard backbones in the final revision to empirically demonstrate the cross-backbone generalization.
> > >
> > > We hope that these clarifications address your concerns. Once again, thank you for the time and effort you dedicated to reviewing the paper.

---

### Official Review · Reviewer_a5JJ · 2026-03-10

**Soundness:** 2
**Presentation:** 2
**Significance:** 2
**Originality:** 2
**Overall Recommendation:** 4
**Confidence:** 1

**Summary:**

This paper proposes PATRA, a framework designed to improve Time Series Question Answering (TSQA) with large language models. The authors identify two main limitations in existing approaches: (1) shallow alignment between text and time-series signals, and (2) optimization imbalance across heterogeneous TSQA tasks. To address these issues, PATRA introduces two key components: a pattern-aware alignment mechanism, which decomposes time-series representations into structural patterns (e.g., trend and seasonality) and aligns them with textual semantics, and an RL-augmented training paradigm that uses task-aware rewards to balance optimization across different task types and improve reasoning ability.

**Compliance With Llm Reviewing Policy:**

Affirmed.

**Key Questions For Authors:**

Seeing weaknesses

**Limitations:**

Seeing weaknesses

**Strengths And Weaknesses:**

Strengths:

1.Clear motivation: The paper clearly identifies two genuine issues in TS reasoning

2. Reasonable architecture design: The pattern-aware alignment idea is intuitive and reasonable.

3. This paper is well writen and easy to follow.

Weaknesses:

1. Limited novelty in alignment method: The pattern decomposition approach is relatively straightforward. It mainly consists of:average pooling for trend and residual for seasonality. This is essentially a simple time-series decomposition applied in embedding space. Besides,  RL component is not very novel. It seems that the RL method combines format reward, task reward and GRPO, which is similar reward structures appear in DeepSeek R1.

2.Comparison baselines are somewhat weak.I'd like to see how more advanced models perform on this task, such as GPT-5.2 / Gemini 3 Pro.

---

> ### Author Rebuttal · Authors · 2026-03-30
>
> We sincerely thank the reviewer for the insightful and constructive feedback. We particularly appreciate the comments on the novelty of the decomposition and RL components, as they help clarify the core design principles and contributions of our method. Below, we address each concern.
>
> ### Q1. The novelty of the decomposition and RL components appears limited.
>
> Response:
> We clarify that the novelty of our method lies not in more complex operators or RL algorithms, but in how they are designed to address cross-modal alignment and task heterogeneity in TSQA.
>
> 1. Decomposition design.
>
> While simple in form, the key novelty lies in **performing decomposition in the LLM latent space for pattern-level alignment**. Unlike conventional pipelines that decompose in raw space and project independently, our design keeps all components in a **unified semantic space**, avoiding representation misalignment. This enables **deep alignment** between time series tokens and language tokens, shifting from sequence-level modeling to pattern-level alignment.
>
> This differs from existing approaches that either (i) perform decomposition followed by independent embedding, or (ii) rely on implicit attention without explicit pattern disentanglement. In contrast, PATRA explicitly **enables pattern-level cross-modal alignment within a unified semantic space**.
>
> Empirically, Appendix G shows strong consistency with real trends (Pearson = **0.986**), and Table 5 shows significant degradation when removing this design.
>
> 1. RL component.
>
> We agree that the underlying optimizer follows existing frameworks. Our contribution lies in **a balanced task reward** design for TSQA.
>
> Due to task heterogeneity (e.g., classification vs. generative reasoning), standard reward tends to over-optimize easier tasks, leading to biased learning and insufficient reasoning capability. Our reward maps heterogeneous signals into a unified space, thereby modifying their relative contributions during optimization and **encouraging the model to allocate sufficient capacity to harder reasoning tasks**.
>
> **This effectively reshapes the optimization, preventing collapse to easier tasks and promoting balanced learning across task types**. As shown in Appendix E.2.2 (Table 6), removing this mechanism leads to **~16%** performance drop, indicating effects beyond simple scaling.
>
> Overall, our contribution is a problem-driven design that combines **pattern-level deep alignment** and **balanced task reward** to **enable effective cross-modal reasoning** in TSQA.
>
> ### Q2. Stronger baselines (e.g., GPT-5.2 / Gemini 3 Pro) should be included.
>
> Response:
> Thank you for this valuable suggestion. We agree that evaluating against stronger LLMs is important. During rebuttal, we include **GPT-5.2** and **Gemini 3 Pro** (*additional Table 1 and Table 2*, https://anonymous.4open.science/r/PATRA-38E7/Rebuttal.md).
>
> From the results, we observe:
>
> - PATRA remains comparable to strong proprietary models such as GPT-5.2 and Gemini-3-Pro, despite using a much smaller model (**7B vs. significantly larger-scale proprietary models**).
> - More importantly, PATRA consistently achieves substantially higher generation quality across all tasks, indicating better alignment with target answers.
>
> In addition, **OOD results** (*additional Table 2,*  https://anonymous.4open.science/r/PATRA-38E7/Rebuttal.md) further highlight the advantage of our approach, PATRA maintains significantly stronger and more stable performance across different settings, demonstrating better generalization ability.
>
> These results suggest that, while large models benefit from scale, incorporating structured time series priors and pattern-aware alignment is particularly beneficial for TSQA, where reasoning depends on explicit time series priors rather than purely increasing model size.

---

> > ### Author Rebuttal · Reviewer_a5JJ · 2026-04-04
> >
> > I believe the author addressed my concerns, and I maintain my positive rating.

---

> > > ### Author Response · Authors · 2026-04-04
> > >
> > > We sincerely thank the reviewer a5JJ for the positive feedback and for recognizing that our responses addressed the concerns. We truly appreciate your time, support, and constructive comments.

---

### Official Review · Reviewer_g28N · 2026-03-13

**Soundness:** 3
**Presentation:** 3
**Significance:** 3
**Originality:** 3
**Overall Recommendation:** 4
**Confidence:** 4

**Summary:**

This paper proposes PATRA, a framework designed to integrate temporal features with LLMs for advanced Time Series Question Answering (TSQA). To address limitations in shallow multimodal alignment and task optimization imbalances, the authors introduce a Pattern-Aware Alignment mechanism that decomposes time series into full, trend, and seasonal components in the latent space, explicitly aligning them with text queries. Furthermore, the framework employs a two-stage training paradigm consisting of SFT followed by GRPO, using a Task-Aware Balanced Reward. Experimental results demonstrate that PATRA achieves SOTA performance on the TSQA benchmark and shows zero-shot out-of-domain generalization on MTBench.

**Compliance With Llm Reviewing Policy:**

Affirmed.

**Key Questions For Authors:**

See weaknesses above.

**Limitations:**

Yes

**Strengths And Weaknesses:**

Strengths:

- Development of better LLM-based systems in the time-series domain is necessary and this paper makes an effort in this direction which is very valuable and timely.
- Patten Aware Alignment seems an innovative and novel approach.
- Showing effectiveness of RL training in time-series is a valuable finding.
- Empirical results are impressive across TSQA. I also appreciate the study on generalization of the proposed method.
- Experiments, and ablations are comprehensive. Paper is easy to understand and well-written.

Weaknesses:

- Why were relevant prior works such as TimeOmni-1, and Time-R1 excluded from the baseline comparisons (also cited in related work)?
- I have a concern regarding the justification for the latent decomposition mechanism. Specifically, extracting the trend by applying average pooling directly in the latent space (Eq 5) is unintuitive since standard time series decomposition relies on temporal moving averages. There needs to be more discussion and justification for this.
- Minor: 'W/ Single-Pattern Alignment' baseline (Table 5) needs to be defined properly, currently it is a bit unclear which pattern (full, trend, or seasonality) was used.
- Does the Pattern-Aware Alignment mechanism explicitly model cross-channel dependencies between M variables, or does the concatenation and subsequent Multi-Head Attention treat them independently, potentially losing critical multivariate correlations required for reasoning tasks?
- Currently, the backbone model for PATRA is Qwen2.5-7B, but do you think it easily scales to other models? Do authors have such a study (I may have missed)?

---

> ### Author Rebuttal · Authors · 2026-03-30
>
> We sincerely thank the reviewer for the thoughtful and constructive feedback.
> We appreciate the insightful comments regarding baseline coverage, decomposition design, and model architecture, which help clarify key aspects of our method and improve the presentation of our work. Below, we address each concern in detail.
>
> ### Q1. Why were recent works such as TimeOmni-1 and Time-R1 not included in the baseline comparison?
>
> Response:
> Thank you for pointing out these relevant works. They were not included in the original submission due to availability constraints. Specifically, **TimeOmni-1** was not publicly available at the time of submission. We have now incorporated it during rebuttal in *additional Table 1 and Table 2* (https://anonymous.4open.science/r/PATRA-38E7/Rebuttal.md).
>
> From the updated results, we observe:
>
> - **Time-R1** checkpoints are still unavailable, preventing reproducible comparison. We also added **ITFormer-7B** to strengthen baselines. Overall, results consistently show that the gains are robust across diverse baselines and reflect the intrinsic effectiveness of PATRA, rather than specific model choices.
> - PATRA-7B consistently outperforms TimeOmni-1-7B and ITFormer-7B across almost tasks (e.g., **37.93 → 56.03** on Comprehension, **41.93 → 64.69** on Recognition), demonstrating strong and stable improvements across different task types;
> - These gains are consistent across both accuracy and generation quality metrics, indicating that PATRA improves not only prediction correctness but also reasoning quality.
>
> In addition, **OOD results** (*additional Table 2,* https://anonymous.4open.science/r/PATRA-38E7/Rebuttal.md) further strengthen this observation; PATRA maintains significantly better and more stable performance, suggesting stronger generalization ability.
>
> ### Q2. The validity of performing decomposition in the latent space.
>
> Response:
> We agree that classical decomposition is performed in the original space. In our setting, time series tokens are already high-dimensional representations where time series patterns are implicitly encoded. Thus, average pooling acts as **semantic** aggregation, analogous to low-pass filtering that preserves global trends.
>
> More importantly, performing decomposition in the latent space ensures that components **remain in a shared semantic space with text tokens**, which is critical for stable cross-modal alignment. In contrast, decomposing in the raw space followed by independent projection may introduce representation inconsistency, making alignment more difficult.
>
> - We validate this in Appendix G (Fig. 7): the extracted latent trend aligns with the true trend (Pearson **0.986**), and the seasonal component also shows high consistency (**0.936**). This indicates that latent-space decomposition remains empirically consistent with classical approaches.
>
> ### Q3. The Clarification on the “W/ Single-Pattern Alignment” setting in Table 5.
>
> Response:
> Thank you for the careful observation. We agree that this description was unclear in the main text.
>
> - In Table 5, the “*W/ Single-Pattern Alignment*” setting corresponds to **using only the trend patterns**. The purpose of this setup is to evaluate how the model performs when relying on a single pattern without complementary information from other components.
> - The results show that this simplified setting leads to a performance drop, indicating that pattern-aware modeling is necessary to capture more complete time series information. We will explicitly clarify this in the final version to avoid ambiguity.
>
> ### Q4. Are cross-variable dependencies explicitly modeled?
>
> Response:
> Cross-variable dependencies are handled by the LLM rather than the alignment module. The alignment module **focuses on cross-modal mapping, aligning time series patterns with textual semantics**.
>
> - As in Eq. (9), all variables are organized into a unified token sequence and processed by the LLM, whose global attention models inter-variable dependencies. This design allows the alignment module to **focus on representation consistency**, while delegating relational reasoning to the LLM, which is better suited for modeling complex dependencies.
> - Ablation results (Table 5) confirm that removing alignment degrades performance.
>
> ### Q5. Can PATRA generalize to other LLM backbones?
>
> Response:
> Yes. PATRA adopts a **non-intrusive design** where the alignment module is decoupled from the LLM. It converts time series features into tokens and injects them via placeholders, without modifying the backbone.
>
> - This **makes it compatible with various architectures** (e.g., DeepSeek/LLaMA), following a general “external module + LLM” paradigm widely used in multimodal systems. In practice, adapting to a new backbone may require lightweight alignment tuning, after which the model can be directly applied.

---

> > ### Author Rebuttal · Reviewer_g28N · 2026-04-01
> >
> > Thanks for the detailed rebuttal. This answers most of my questions, but I still have follow-up on this comment - "Currently, the backbone model for PATRA is Qwen2.5-7B, but do you think it easily scales to other models? Do authors have such a study (I may have missed)?".
> >
> > I agree that PATRA definitely is compatible with various architectures (e.g., DeepSeek/LLaMA), however, my concern with performance implication of using different models. Current results are using `Qwen2.5-7B`, but if we change underlying model with maybe similar size, but different family model as backbone, do the results/claims stay the same? My question mainly was do the authors have insights or some preliminary results/study on this?

---

> > > ### Author Response · Authors · 2026-04-02
> > >
> > > Thank you for your insightful question. We fully agree that testing with various backbones is an important aspect to ensure the robustness of the method across different architectures.
> > >
> > > Due to the limited time in the rebuttal phase, we were not able to conduct a full cross-backbone evaluation. However, we will include additional analysis and, where possible, further empirical validation in the final version to better address this concern.
> > >
> > > We note that recent TSQA models such as **TimeOmni-1[1]**, **ITFormer[2]**, and **ChatTS[3]** are also built upon similar LLM backbones (Qwen2.5-7B or larger-scale variants). This suggests that performance differences in this line of work are largely influenced by how time series information is integrated and how training is conducted, rather than the choice of backbone itself.
> > >
> > > And PATRA is designed to be **backbone-agnostic**, as both the pattern-level alignment and the task-aware reward operate at the representation and training level without modifying the internal architecture of the LLM. As a result, the improvements mainly stem from how time series information is structured and how optimization is balanced, rather than from backbone-specific properties.
> > >
> > > Therefore, while absolute performance may vary across model families, the relative improvements over corresponding baselines are expected to remain consistent. Our ablation results (Table 5) further suggest that the performance gains are driven by the overall framework design, rather than any specific component or configuration tied to a particular backbone.
> > >
> > > [1]  Timeomni-1: Incentivizing complex reasoning with time series in large language models, Guan, Tong, et al., ICLR, 2026.
> > >
> > > [2] ITFormer: Bridging Time Series and Natural Language for Multi-Modal QA with Large-Scale Multitask Dataset, Wang, Yilin, et al., ICML, 2025.
> > >
> > > [3] Chatts: Aligning time series with llms via synthetic data for enhanced understanding and reasoning, Xie, Zhe, et al., VLDB, 2025.

---

### Decision · Program_Chairs · 2026-04-30

**Decision:**

Accept (regular)

**Comment:**

This paper received overall positive reviews (5, 4, 4, 3). While the reviewers expressed concerns on limited technical novelty---with Reviewer gA2i arguing that the novelty is "not very convincing" and Reviewer a5JJ noting the components are "technically lightweight"---this meta-reviewer found the paper’s empirical success in the domain of Time Series Question Answering (TSQA) to be timely and meaningful. After the rebuttal, the reviewers were convinced by the Pattern-Aware Alignment mechanism. Furthermore, the Task-Aware Balanced Reward and GRPO-based training were shown to effectively stabilize learning across heterogeneous tasks, even performing competitively against strong frontier, proprietary models like GPT-5.2 in specialized reasoning. Although some concerns remain regarding the "engineering-heavy" nature of the contribution, the overall feedback is that the framework’s robust performance and successful handling of task heterogeneity provide a solid, practically meaningful contribution to the field.